# Single-cell RNA sequencing unveils the hidden powers of zebrafish kidney for generating both hematopoiesis and adaptive antiviral immunity

Chongbin Hu[1], Nan Zhang[1], Yun Hong[1], Ruxiu Tie[2], Dongdong Fan[1], Aifu Lin[1], Ye Chen[1,3]*, Li-xin Xiang[1]*, Jian-zhong Shao[1,4]*

[1]College of Life Sciences, Key Laboratory for Cell and Gene Engineering of Zhejiang Province, Zhejiang University, Hangzhou, China; [2]Bone Marrow Transplantation Center, the First Affiliated Hospital, Zhejiang University School of Medicine, Hangzhou, China; [3]Department of Genetic and Metabolic Disease, The Children's Hospital, Zhejiang University School of Medicine, National Clinical Research Center for Child Health, Hangzhou, China; [4]Laboratory for Marine Biology and Biotechnology, Qingdao National Laboratory for Marine Science and Technology, Qingdao, China

*For correspondence:
yechency@zju.edu.cn (YC);
xianglx@zju.edu.cn (L-xinX);
shaojz@zju.edu.cn (J-zhongS)

Competing interest: The authors declare that no competing interests exist.

**Abstract** The vertebrate kidneys play two evolutionary conserved roles in waste excretion and osmoregulation. Besides, the kidney of fish is considered as a functional ortholog of mammalian bone marrow that serves as a hematopoietic hub for generating blood cell lineages and immunological responses. However, knowledge about the properties of kidney hematopoietic cells, and the functionality of the kidney in fish immune systems remains to be elucidated. To this end, our present study generated a comprehensive atlas with 59 hematopoietic stem/progenitor cell (HSPC) and immune-cells types from zebrafish kidneys via single-cell transcriptome profiling analysis. These populations included almost all known cells associated with innate and adaptive immunity, and displayed differential responses to viral infection, indicating their diverse functional roles in antiviral immunity. Remarkably, HSPCs were found to have extensive reactivities to viral infection, and the trained immunity can be effectively induced in certain HSPCs. In addition, the antigen-stimulated adaptive immunity can be fully generated in the kidney, suggesting the kidney acts as a secondary lymphoid organ. These results indicated that the fish kidney is a dual-functional entity with functionalities of both primary and secondary lymphoid organs. Our findings illustrated the unique features of fish immune systems, and highlighted the multifaced biology of kidneys in ancient vertebrates.

## eLife assessment

This study characterizes the composition and immune diversity of the zebrafish kidney, the immune organ equivalent to human bone marrow, with **convincing** single-cell transcriptomic data of hematopoietic cells and immunocytes. The key findings suggest that zebrafish kidney is a secondary lymphatic organ, and that hematopoietic stem cells in zebrafish may exhibit trained immunity, which are the unique features of the fish immune system. This study provides new and **valuable** insights into the antiviral response in teleost fish, which will be of interest to biologists in general, and to immunologists and cancer researchers in particular.

## Introduction

Teleost fish are a group of indispensable model organisms for comprehending the evolutionary history and general principles underlying the design of the vertebrate immune system, since they stand as the earliest living jawed vertebrates known as an important link in vertebrate evolution (*Sunyer, 2013*). Over the recent decades, the field of fish immunology has made significant strides, notably hastening the identification and characterization of functional genes and signaling pathways intricately associated with both piscine innate and adaptive immunities. These advances stem from the progress in genome projects encompassing diverse fish species, and have markedly enriched our comprehension of the molecular underpinnings of fish immunity (*Zhu et al., 2013*; *Volff, 2005*; *Fan et al., 2020*). Some breakthroughs have not only challenged established paradigms about the immune system but have also unveiled novel dimensions of mammalian immunity (*Sunyer, 2013*; *Zhu et al., 2013*). Thus far, it was generally accepted that teleost fish possess the fundamental components of both innate and adaptive immune systems, akin to those seen in humans and other mammalian counterparts (*Rauta et al., 2012*; *Secombes and Wang, 2012*). However, our grasp of the precise architectural layout, cellular coordination, and functional attributes of the fish immune system, particularly the adaptive facet, remains incomplete (*Secombes and Wang, 2012*; *Mokhtar et al., 2023*). Evolutionarily speaking, bony fish represent the earliest living organisms endowed with a rudimentary adaptive immune system encompassing basic molecular and cellular constituents. These encompass immunoglobulins (Igs), antigen-specific receptors (TCR/BCR) driven by the recombination-activating genes, major histocompatibility complex class I and II (MHC-I/II) molecules, as well as T and B lymphocytes. These cellular elements populate the primary and secondary lymphoid organs of fish, such as the thymus and spleen (*Smith et al., 2019*; *Cooper and Alder, 2006*; *Wan et al., 2016*). Notwithstanding, the fish adaptive immune system is characterized by its own set of specializations and distinctive attributes. Notably absent in fishes are the bone marrow, histologically discernible lymph nodes, Peyer's patches, and germinal centers—entities that define primary and secondary lymphoid organs in mammals (*Neely and Flajnik, 2016*; *Matz et al., 2023*). Additionally, fishes lack antibody class-switch recombination activity, albeit retaining the ability to express activation-induced cytidine deaminase (AID) (*Barreto et al., 2005*; *Wakae et al., 2006*). These revelations underscore the substantial diversity inherent in immune systems across distinct vertebrate taxonomic groups. Consequently, teleost fish emerge as invaluable primitive animal models, illuminating previously uncharted events during the emergence and phylogenetic progression of adaptive immune systems. Such endeavors stand to elucidate the distinct principles governing the immunology of ancient vertebrates, thereby furnishing scientific substantiation for a broader comprehension of the evolutionary trajectory of vertebrate immune systems.

In light of these considerations, we have recently generated a comprehensive atlas of immune cell types within the spleen of zebrafish, employing a single-cell transcriptome profiling approach (*Hu et al., 2023*). Our work has entailed the classification of splenic leukocytes into 11 principal categories and the identification of over 50 subset populations within these categories. Moreover, we have delineated the differential responses of various subset cells—both innate and adaptive—to infection by the spring viremia of carp virus (SVCV). Particularly noteworthy was our discovery of hematopoietic stem and progenitor cells (HSPCs) that existed within the spleen. This observation intimates that the fish spleen may well function as a hematopoietic site, transcending its established role as a secondary lymphoid organ. These insights offer fresh perspectives into the intricacies and distinctive attributes of the fish immune system, setting it apart from its mammalian counterparts. Seeking to attain a deeper understanding of the fish immune system, our present study has undertaken a further exploration of immune cell types and their functional characteristics within the kidney of zebrafish. This investigation places special emphasis on the hematopoietic activity of kidney HSPCs and the adaptive immune responses of kidney immune cells toward viral infections. As a result, our classification efforts have yielded 13 categories of cells from kidney leukocyte preparations, unearthing 59 potential subset cells spanning HSPCs and immune cell-associated categories. The observed subset populations manifest disparate reactions to SVCV infection. Most notably, some HSPC subsets exhibit a robust proliferative response to viral infection, capable of inducing trained immunity. Furthermore, we have successfully identified a comprehensive array of cells—encompassing diverse antigen-presenting cells (APCs) as well as effector T and B cells—integral for the full activation of adaptive immunity within the kidney. These adaptive immune-associated cells display marked responsiveness to SVCV infection

and antigenic stimulation. Evidently, the complete process of adaptive immunity unfolds within the kidney, as evidenced by the substantial expansion, activation, and somatic hypermutation (SHM) of antigen-specific T and B cells upon exposure to cognate antigens. These results strongly affirmed the kidney's dual role as a lymphoid organ, boasting both hematopoietic functionality and adaptive immune reactivity in adult fish. Consequently, our findings cast the fish kidney in the role of a secondary lymphoid organ, supplementing its recognized status as a primary lymphoid organ. This advancement significantly improved our current comprehension of fish hematopoietic and immune systems, thus advancing our insights into the multifaceted biology of the kidney in ancient vertebrates.

## Results
### Categories of immune-associated cells in the kidney

In order to profile distinct immune-cell categories within the kidneys, leukocytes were extracted from zebrafish kidney tissues. The experimental groups consisted of fish treated with PBS (control), fish infected with SVCV (infected), and fish that underwent both vaccination with inactivated SVCV and subsequent SVCV infection (vaccinated+infected). Each sample comprised leukocytes from 60 fish and was subjected to scRNA-seq using the 10x Chromium platform. Following sequencing, the control group exhibited mean reads/median genes per cell of 34,501/1090, the infected group showed 29,706/1383, while the vaccinated+infected group displayed 37,611/1379. The total number of detected genes in leukocyte samples from the control, infected, and vaccinated+infected groups were 26,345, 26,865, and 25,342, respectively. Employing an unsupervised cluster detection algorithm (SEURAT), we identified 13 distinct clusters of cells exhibiting similar gene expression profiles (*Figure 1A*). Moreover, specific genes were pinpointed within each cluster. Notably, clear demarcation boundaries were evident for each cluster in the heatmap, while dot plot and violin plot analyses indicated that a PCA-based cell separation approach was indeed favorable for our investigation (*Figure 1B and C*, *Figure 1—figure supplement 2*).

The 13 clusters were classified as neutrophils, macrophage/myeloid cells, B cells, T/NK cells, and three categories of HSPCs (HSPCs 1–3), along with renal cells (kidney multiciliated cells, kidney mucin cells, kidney distal tubule, and proximal tubule cells) (*Figure 1A–C*). Clusters 0, 2, and 9 were identified as neutrophils based on their expression of lineage marker genes including *mpx*, *lyz*, *cpa5*, *cd11b*, *adgrg3*, and *nccrp1* (*Figure 1C and D*). Notably, cluster 0 exhibited high expression of *scpp8*, *mmp13a.1*, *mmp9*, *tlr5b*, and *gpr84*, while cluster 2 was actively expressing interferon-inducible genes like *zgc:152791* (*ifi27.2*, interferon alpha inducible protein 27.2) and *si:dkey-188i13.6* (*ifi27.3*, interferon alpha inducible protein 27.3). Cluster 9, on the other hand, displayed elevated expression of *hmgn2*, *hmgb2a*, and *top2a*. The functional implications of these genes in neutrophils necessitate further exploration. Cluster 1 was identified as a macrophage/myeloid population due to the expression of lineage-specific genes including *mpeg1.1*, *ctss2.2*, *mpeg1.2*, *grn1*, *marco,* and *grna*. Additionally, cluster 1 exhibited expression of *s100a10b*, *lgals3bpb*, and *c1qb*, implicated in classical complement pathway activation (*Figure 1C and E*). Cluster 3 represented B cells, expressing lineage marker genes like *ighm*, *ighd*, *ighz*, *igl3v5*, *cd79a*, *cd79b* and *cd37* (*Figure 1C and F*). Consistent with the existence of Mpeg1.1⁺ B cells in zebrafish kidney, gut, skin, and spleen tissues (*Ferrero et al., 2020*), this cluster displayed expression of *mpeg1.1*. Notably, cluster 3 also highly expressed *igic1s1*, *zgc:194275*, *igl1c3*, and *cxcr4a*, the latter harbors major chemotactic activity in lymphocytes. Clusters 4 and 6 corresponded to HSPCs 1 and HSPCs 2 populations, marked by the expression of pertinent signature genes like *nanos1*, *spi2*, *meis1b*, *foxp4*, *csf1rb*, *gtpbp4*, *rpl5b*, *rplp2l*, *rpl18*, *rps7*, *rsl1d1* in cluster 4, and *mki67*, *mcm5*, *dut*, *npm1a*, *snu13b*, *cnbpa* in cluster 6 (*Figure 1C, I and J*; *Rubin et al., 2022*). Cluster 5 represented T and NK cells, expressing T-cell signature genes including *trac*, *trbc*, *cd3ζ*, *lck*, *zap70*, *cd4-1*, *cd8a*, *runx3*, *il2rb*, *bcl11ba* and *il7r*, as well as NK lineage genes like *nkl.2*, *cd56*, *cd94*, *cd27* and *zbtb32* (*Figure 1C, G and H*). Additionally, cluster 5 also highly expressed *ccl34b.4*, *ccl33.3*, *ccl36.1*, *nk-lysin*, *sla2*, *tnfrsf18*, and *tnfrsf9*, the latter contributes to the clonal expansion, survival, and development of T cells (*Philipson et al., 2020*; *van Asten et al., 2021*). Cluster 8 denoted the HSPCs 3 population, expressing *zfpm1*, *meis1b*, *gata1a*, *zeb2a*, *tal1*, *mef2aa*, *mpl*, *pbx1a* and *hdr* marker genes (*Figure 1C and K*). This HSPC population also actively expressed *hsp70* (*hsp70.1*, *hsp70l*, *hsp70.3*), *itga2b*, *thbs1b*, *hbaa1*, *hbaa2*, *hbba1*, *hbba1.1*, and *rgcc*, among which *hsp70* members are essential for the differentiation of hematopoietic progenitors into

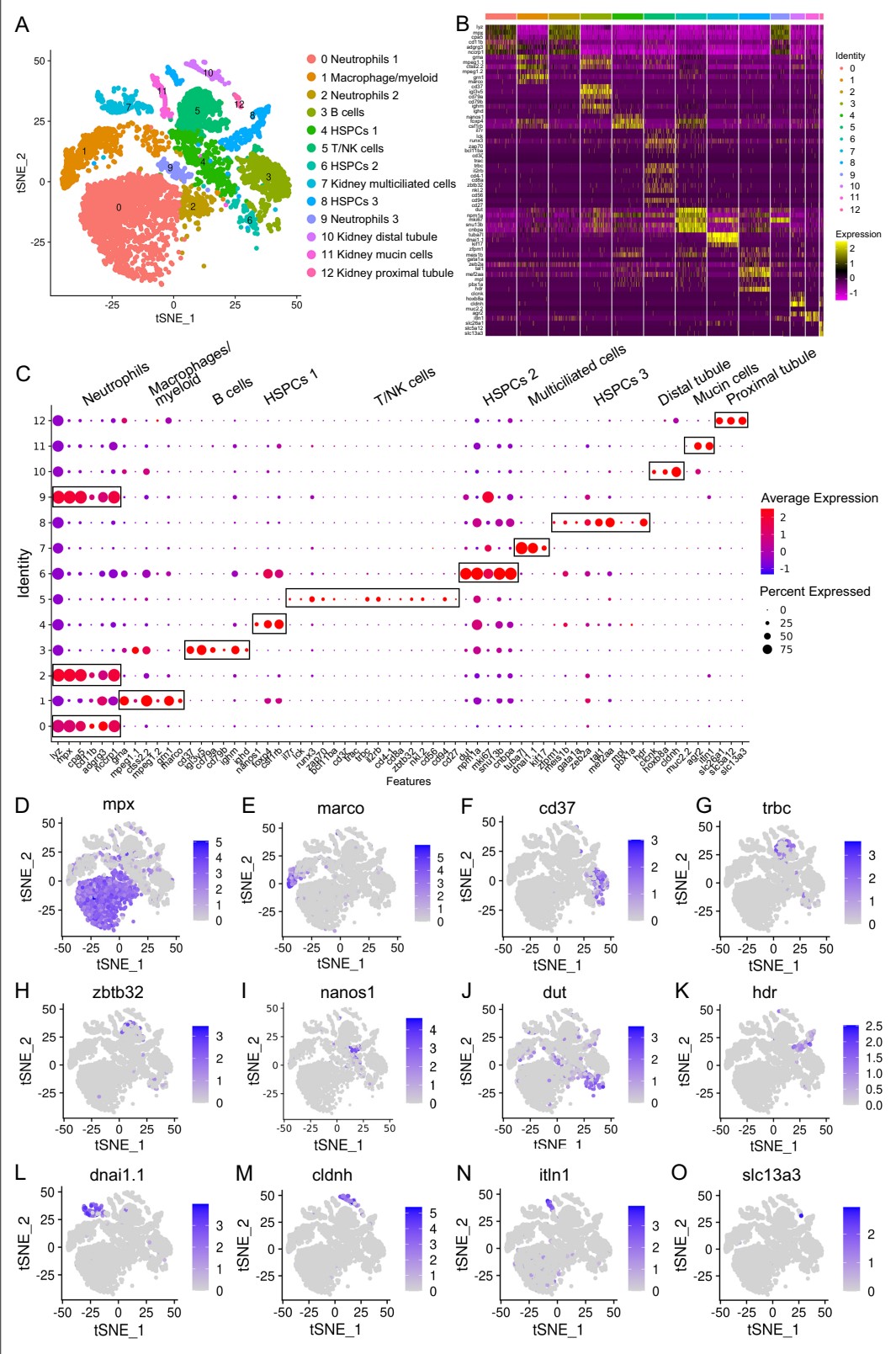

**Figure 1.** Categorization of the hematopoietic stem/progenitor cells (HSPCs) and immune cell types in zebrafish kidney. (**A**) Nonlinear t-SNE clustering visually categorized populations of zebrafish kidney leukocytes. Principal component analysis (PCA) was performed following expression data normalization, with subsequent PCA dimension reduction to streamline variables. The cell clustering algorithm based on graph theory was then

*Figure 1 continued on next page*

*Figure 1 continued*

employed. (**B**) Heatmap displaying marker genes specific to each cluster. Columns depict distinct cell subtypes, while rows correspond to genes. Expression levels are color-coded, with intense yellow indicating high expression and deep purple representing low expression. (**C**) Dot plots illustrating marker gene expression levels and the respective percentages of cluster-expressing cells. (**D–O**) t-SNE results depicting molecular marker distribution for immune and kidney cell types. Each dot signifies a cell, with dark dots indicating elevated expression of the marker gene in a given cell. Notably, Neutrophil marker gene in (**D**), Macrophage/myeloid cells marker gene in (**E**), B-cells marker gene in (**F**), T-cells marker gene in (**G**), NK-cells marker gene in (**H**), HSPCs 1 marker gene in (**I**), HSPCs 2 marker gene in (**J**), HSPCs 3 marker gene in (**K**), kidney multiciliated cells marker gene in (**L**), kidney distal tubule cells marker gene in (**M**), kidney mucin cells marker gene in (**N**), and kidney proximal tubule cells marker genes in (**O**).

The online version of this article includes the following figure supplement(s) for figure 1:

**Figure supplement 1.** Single-cell sequencing quality control.

**Figure supplement 2.** Violin plots showing the expression of marker genes for per immune cell and renal cell categories at the single cell level.

---

erythroids (***Ribeil et al., 2007***). Clusters 7, 10, 11, and 12, characterized by lineage-specific genes like *tuba7l, dnai1.1, kif17; clcnk, hoxb8a, cldnh; muc2.2, agr2, itln1*; and *slc26a1, slc5a12, slc13a3* respectively, represented kidney multiciliated cells, kidney distal tubule cells, kidney mucin cells, and kidney proximal tubule cells (***Figure 1C and L–O***; ***Tang et al., 2017***). These renal cells are remnants in the preparation of leukocytes. It's noteworthy that cluster 11 exhibited high expression of genes like *zmp:0000001323, CABZ01058647.1, si:dkey-203a12.2* (*spink2.7*), *spink2.5, spink2.10*, and *spink2.4*, resembling a unique serpin-secreting cell population recently identified in zebrafish spleen tissues (***Hu et al., 2023***). The detailed information about the marker genes used to classify each cluster of cells is provided in the ***Supplementary files 1 and 2***.

## Characterization of kidney HSPCs

To achieve a more comprehensive understanding of kidney hematopoietic cell populations, sub clustering was conducted exclusively on the three HSPC clusters depicted in ***Figure 1***. This analysis unveiled 11 enrichments (designated as H0–H10) (***Figure 2A***). To validate our outcomes, we aligned the subset-specific genes of each cluster and the ensuing heatmap indicated a distinct categorization for each cluster (***Figure 2B***). Within this context, H0 manifested as a Nanos1$^+$Spi2$^+$ phenotypic cluster expressing signature genes *nanos1* and *spi2*, alongside specific expressions of *sla2, csf1rb, rn7sk, sik1,* and *irf1b*, forming a collection of potential new marker genes (***Figure 2A–C***). Meanwhile, H1 was identified as a group of cycling HSPCs, characterized by the expression of the marker gene *mki67*. This cluster also exhibited pronounced expression of *hist1h4l.12, tubb2b, hmgb2a, hmgb2b, zgc:110425, grn1, BX640512.2,* and *si:ch73-281n10.2* genes (***Figure 2A–C***). It's noteworthy that the presence of H0 and H1 populations has been corroborated in previous research (***Rubin et al., 2022***). H2, H3, and H4 clusters demonstrated high expression of potential signature genes: *blf, hdr, rhag, mef2aa, rgcc, pmp22b, fth1a, prdx2, mych, igfbp1a* in H2; *CR318588.2, zgc:152791, si:busm1-266f07.2, CR318588.3, CR936442.1, tnfrsf9a, tat* in H3; and *BX640512.2, grn1, BX323596.2, cxcl11.1, marcksl1b, lgals2a, txn, psma3, atp5mc1* in H4 (***Figure 2A–C***), respectively. The functional characteristics and developmental potential of these three clusters require further elucidation. H5 was categorized as a group of NK progenitors, exhibiting high expression of *gata3, nkl.3, nkl.4, ccl34b.4, CR388164.2, ccl33.3, dusp2, BX005105.1, icn, bhlhe40,* and *coro1a* (***Figure 2A–C***). Cells within H6 represented myeloid progenitors, showing extensive expression of *mmp13a.1, si:ch1073-429i10.1, il1b, cpa5, mpx, timp2b, BX908782.2, npsn, lect2l* and *lyz* marker genes (***Figure 2A–C***). H7 and H8 were identified as erythrocyte and lymphoid progenitor populations, denoted by their expression of *hbba1.1, hbaa1, hbba1, si:ch211-5k11.8, hbba2, hbaa2, si:ch211-103n10.5, cahz, hemgn* in H7, and *pbx1a, pdpk1a, rag1, rag2, cxcr4b, h2afx, ccr9a, sid1, zgc:77784, nme3, kdm6bb* in H8, respectively (***Figure 2A–C***). H9 and H10 clusters were discerned as endothelial and thrombocyte progenitors due to their expression of marker genes such as *cdh5, kdrl, pecam1, igfbp5b, rbp4, cxcl18b, cxcl12a, selenop, jun, fosl1a, jdp2b, fosab, si:ch73-335l21.4* in H9, and *itga2b, fn1b, thbs1b, sele, aldob, apln, tpm4a, hbegfb, cald1a, fgl2a, pim2, atp2b1a* in H10 (***Figure 2A–C***). The endothelial progenitors were also found in the bone marrow of mouse, and they are developmentally associated with a subset of

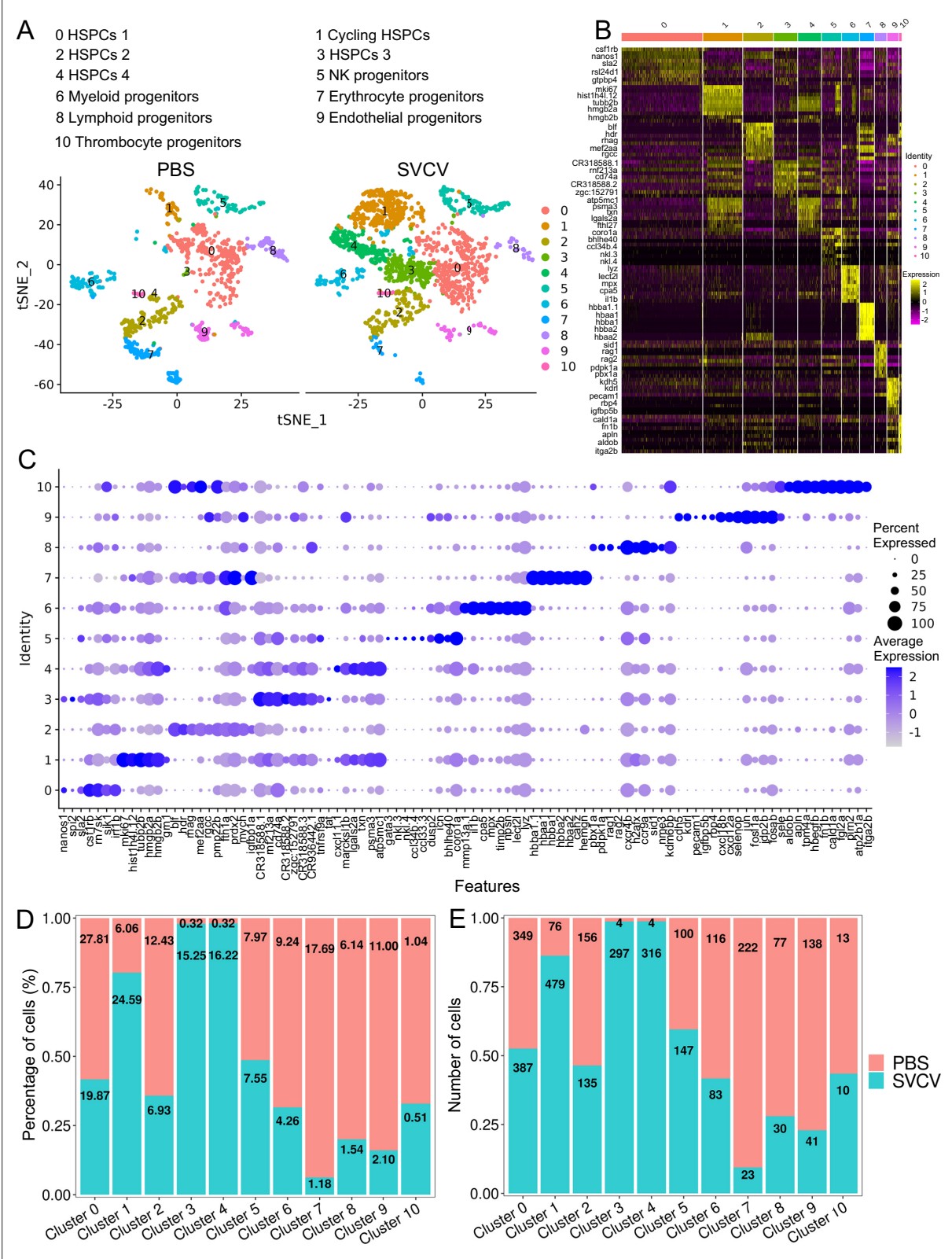

**Figure 2.** Hematopoietic stem/progenitor cells (HSPCs) subtype analysis based on single-cell gene expression. (**A**) Graph-based clustering revealed 11 subclusters of HSPCs within both the control group (PBS) and the spring viremia of carp virus (SVCV)-infected group. (**B**) Heatmap presenting marker gene expression for each HSPC subcluster. (**C**) Dot plots of differentially expressed selected markers across HSPCs subtypes. Dot size indicates the percentage of cells within a population expressing a specific marker, while dot color indicates average expression. (**D**) Histogram demonstrating

*Figure 2 continued on next page*

*Figure 2 continued*

differences in HSPC subset ratios between the control group (PBS) and SVCV-infected group. (**E**) Histogram illustrating discrepancies in HSPC numbers across clusters between the control group and the SVCV-infected group.

CD34$^+$ hematopoietic stem cells, and can differentiate ex vivo to an endothelial phenotype (***Asahara et al., 1997***; ***Shi et al., 1998***). The detailed information about the signature genes used to characterize each cluster of HSPCs is provided in the ***Supplementary files 1 and 3***.

Subsequently, pseudotime trajectory analysis revealed that a significant portion of HSPCs were positioned within a primary trajectory characterized by five bifurcations and five stages (***Figure 3A–C***). The endothelial (H9) and thrombocyte (H10) progenitors were situated toward the trajectory's origin. This arrangement partially validated the constructed trajectory (***Figure 3C***). H4 cells were positioned towards the trajectory's terminus (***Figure 3C***). H2 cells, along with erythrocyte (H7) and lymphoid (H8) progenitors, were positioned towards the trajectory's midpoint. H0 and H3 populations, along with NK (H5) and myeloid (H6) progenitors, were located towards both the origin and midpoint of the trajectory. Furthermore, an analysis of the pseudotime dynamics of significantly changed genes in the 11 HSPC clusters revealed three modules delineated by their pseudotemporal expression patterns (***Figure 3D and E***). In addition, RNA velocity analysis showed the differentiation potential of H0, H1, H2, H3, and H4 subclusters into myeloid, thrombocyte/erythrocyte, and lymphoid progenitors (***Figure 3F***). These findings collectively suggested that H0-H4 clusters potentially represent a group of stem and early progenitors that lead to late progenitors responsible for differentiation into erythroid, myeloid, and lymphocyte lineages. Notably, we recently detected the presence of HSPCs in zebrafish spleen tissues, indicating the spleen's role as a hematopoietic organ in fish (***Hu et al., 2023***). To compare HSPC characteristics between the kidney and spleen, we integrated scRNA-seq data from both organs and subjected the merged dataset to cell classification. This assessment demonstrated that the newly generated atlas shared overall consistency with the kidney and spleen, merging proportional HSPCs and immune cells from the two organs to varying extents (***Figure 4A***). These observations underscored the presence of substantial overlapping HSPCs and immune cell types in both the kidney and spleen. Furthermore, pseudotime trajectory analysis highlighted similar cell differentiation trajectories shared between the kidney and spleen (***Figure 4B–E***). Pseudotime dynamics of significantly altered genes across all categories were analyzed and organized into three modules based on their pseudotemporal expression patterns (***Figure 4F and G***). In aggregate, our findings reveal overall similarities in HSCPs' composition and differentiation potentials between the kidney and spleen, suggesting analogous functionalities in hematopoiesis and immunity across both organs. Nevertheless, the precise correlation between the fish kidney and spleen requires further clarification.

## T and NK subsets in kidney

Subdividing the T/NK category in ***Figure 1*** led us to observe 23 subset clusters (designated as T0–T22) enriched within the T/NK population through iterative subclustering (***Figure 5A***). To validate the classification results, we assessed the most variably expressed genes among cell clusters, and the heatmap demonstrated effective classification for each cluster (***Figure 5B***). Subsequently, we analyzed the DEGs within the 23 clusters. Notably, clusters T5, T17, and T18 were identified as NKT subsets due to their widespread expression of NKT lineage genes, including *cd3ζ, trbc, cd8a/cd4-1, zbtb32, nkl.2, il2rb,* and *cd94* (***Figure 5C***). These NKT subset clusters exhibited substantial heterogeneity, as evident from their distinct transcriptional profiles, including the high expression of *vmp1, BX649554.1, nkl.2, stim1a, ccl36.1, CABZ01044746* in T5; *gata3, cpa5, tppp, si:dkey-78l4.7, si:dkey-78l4.4, cfbl, si:dkey-21e2.3.1, BX322787.1, si:dkey-9i23.4, si:dkey-78l4.6* in T17; and *ccl39.7, ccl39.6, nfil3−4, ccl35.2* in T18. Cluster T0 represented the CD3$^{low}$CD4$^{low}$CD8$^{low}$ T subset due to its low expression of *cd3ζ, cd4-1,* and *cd8a*. This subset likely represents an early developmental population of double positive (DP) T cells (***Figure 5C***). Clusters T1 and T2 were identified as CD8$^+$-nkl.2 subsets exhibiting cytotoxic activity, as evidenced by the expression of *cd8a* and *nkl.2* (***Figure 5C***). T3 was classified as a Th1 subset based on its expression of Th1-specific genes, including *cd4-1, ifng1, il2, cxcr3.1.1, tnfa,* and *tnfb*. Additionally, T3 expressed the cytotoxic T cell-related *gzmk* gene, suggesting its potential for both helper and cytotoxic activities (***Figure 5C***). T4 was designated as a CD4$^-$CD8$^-$-nkl.4 subset due to its high expression of *nkl.4* and *trbc*, coupled with the absence of *cd4/cd8* expression and low *cd3ζ* expression. Continuing the narrative, clusters T6, T7, T10, T13, T19, T20, and T21 were categorized

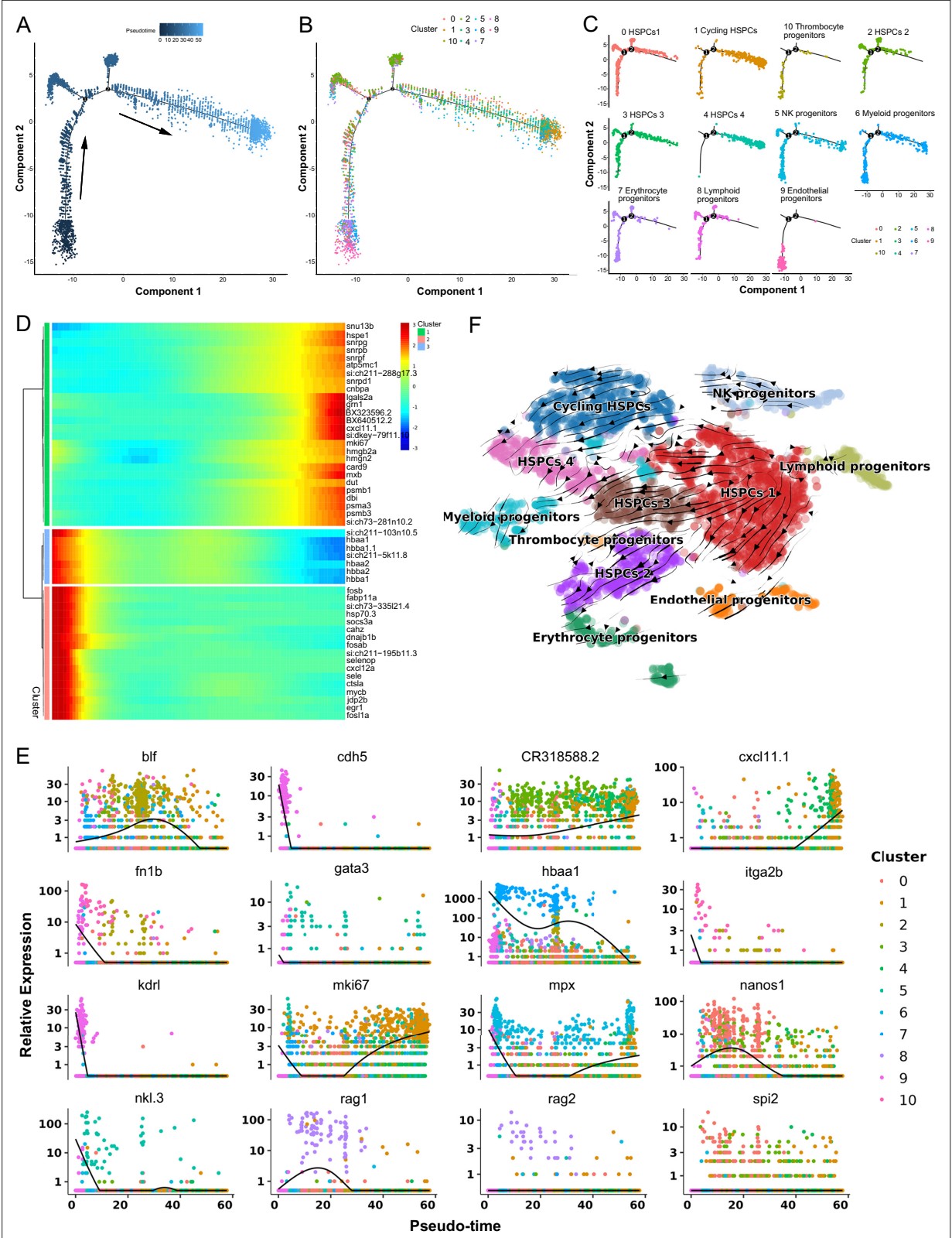

**Figure 3.** Pseudotemporal and RNA velocity analyses of zebrafish kidney hematopoietic stem/progenitor cells (HSPCs) development. (**A**) Pseudotime single-cell trajectory of HSPCs reconstructed using Monocle2. Pseudotime is represented in a gradient from dark to light blue, with its starting point indicated. (**B–C**) Distribution of the 11 HSPC subsets within the trajectory. (**D**) Pseudotemporal heatmap demonstrating gene expression dynamics of significant marker genes. Genes (rows) were grouped into three modules, while cells (columns) were ordered according to pseudotime. (**E**) Gene

*Figure 3 continued on next page*

*Figure 3 continued*
expression dynamics over pseudotime computed in Monocle2, color-coded by cluster assignment for 16 genes linked to HSPCs development. (**F**) Single cell velocities of the HSPCs subclusters are visualized as streamlines in a t-distributed stochastic neighbor embedding (t-SNE) plot. Black arrows indicate direction and thickness indicates speed along the HSPCs development trajectory.

as NK subsets due to their expression of NK signature genes, including *zbtb32*, *nkl.2*, *il2rb*, and *cd94* (**Figure 5C**). Among these, T19 exhibited no expression of *cd11b* or *cd27*, T7 expressed both *cd11b* and *cd27*, and T6, T10, T13, T20, and T21 expressed *cd11b* but not *cd27* (**Figure 5C**). These observations suggested that T19, T7, and T6/10/13/20/21 represented CD11b⁻CD27⁻, CD11b⁺CD27⁺, and CD11b⁺CD27⁻ subset NK cells, demonstrating high differentiation potential, cytokine secretion, and cytolytic functions as observed in humans (**Fu et al., 2011**). Notably, each NK subset displayed distinct gene expression profiles, including the expression patterns of *her6*, *txn*, *id2a*, *mef2cb*, *zgc:162939* in T6; *hsp70.3*, *nkl.3*, *dicp1.1*, *nkl.4*, *ccl33.3* in T7; *cxcr4a*, *id1*, *phlda2*, *gnb5b*, *rasgef1ba*, *lgals2a*, *ctsbb*, *BX649442.1*, *enpp1* in T10; *nkl.2*, *nkl.3*, *nkl.4*, *fa2h*, *flt3* in T13; and *ctss1*, *eomesa*, *mafbb*, *mafa*, *si:ch73−180n10.1*, *CR388164.3*, *CR388164.2* in T19. Additionally, T20 and T21 cells specifically expressed *grn2* and *diabloa*, respectively. These findings underscored the heterogeneity and functional diversity of zebrafish NK subsets. Clusters T8, T9, and T15 were identified as a group of CD8⁺ CTL subsets, expressing CD8⁺ CTL-specific genes, including *cd8a*, *cd8b*, *gzma*, *gzmk*, *ifng1*, and *tnfb* (**Figure 5C**). T11 was labeled as a γδ T subset due to its high expression of *trgc*, *trdc*, *sox13*, and *cd3ζ*, combined with low expression of *cd4-1* and *cd8a*. This γδ T subset also exhibited high expression of *il2*, a pivotal cytokine gene involved in diverse immune activities (**Figure 5C**). T12 was classified as a Th17 subset due to its expression of lineage-specific genes, including *roraa*, *stat3*, *il17a/f3*, *il21*, *cd4-1*, *ccr6a*, *rorc*, *cd3ζ*, *il22*, *tgfb1a*, and *il17a/f1*. T14 represented a distinctive CD4⁺IgM⁺ subset characterized by the concurrent expression of *ighm* and *cd4-2.2*. Cluster T16 was designated as a regulatory T (Treg) subset based on the expression of *foxp3a*, *foxp3b*, *ctla4*, *cd4-1*, and *il17ra1a* marker genes (**Figure 5C**). T21 was indicative of a Th2-like subset, demonstrated by its expression of Th2-associated genes, including *cd3ζ*, *il4*, *gata3*, *il13*, *stat6*, *batf*, and *cxcr4b*. Notably, this subset exhibited elevated expression of *faslg*, a pivotal apoptotic inducer contributing to T cell cytotoxicity (**Figure 5C**, Figure 9D). The detailed information about the marker genes used to characterize each subset of T and NK cells is provided in the **Supplementary files 1 and 4**.

## B cell subsets in the kidney

Nine subclusters of B cells (designated as B0–B8) were classified through unsupervised clustering of all B cells, based on their expression of distinct B-subset signature genes (**Figure 6A and B**). B0 and B3 were designated as immature/mature B cells (im/mat. B) due to their expression of *gata1a* and *zfpm1* genes (**Figure 6A and B**). B1 was categorized as a canonical IgM⁺IgD⁺ B subset, characterized by its high expression of *ighm* and *ighd* genes. B2 and B7 were identified as pro/pre B subsets, exhibiting significant expression of *pclaf*, *pdia4*, *pdia6*, *rag1*, and *rag2* genes. B4 likely represented a population of plasma cells that highly expressed *ighm* or *ighz* (**Figure 6A and B**). B5 displayed expression of genes related to germinal-center B cells, including *aicda* (AID), *mki67*, *top2a*, *stmn1a*, *bach2b*, *bcl6aa*, *bcl6ab*, *cd81a*, *cd9a*, *irf8*, *lmo2*, *pclaf*, and *nfkb2*, which exhibited a notable increase following virus infection (**Figure 6A–C**). Cluster B6 comprised activated B cells and plasmablasts (act. B/PBs), actively expressing *dscamb*, *cpa5*, *fcer1g*, *mmp13a.1*, *mpx*, *timp2b*, *npsn*, *il1b*, and *lect2l*. It's worth noting that B6 also expressed *il10* and *tgfb1a/b*, suggesting the potential presence of a regulatory B cell type within this subcluster (**Figure 6A and B**). B8 was recognized as a B1-like subset with heightened immune activities, including chemokine production, based on its expression of *cd5*, *ccl33.3*, *ccl34b.4*, *nkl.4*, *tnfrsf18*, *nkl.2*, *sla2*, and *hmgn3* genes (**Figure 6A and B**). Interestingly, a distinct subset of IgM⁺β⁺ cells was identified within B8 (**Figure 6A and B**). This lineage, known as dual expresser (DE), simultaneously expressing TCR and BCR, has been reported in humans and is also present in the spleen of zebrafish (**Hu et al., 2023**; **Ahmed et al., 2019**). The top 10 subset-specific genes with the greatest significance for each B-cell subset are visualized in a heatmap (**Figure 6—figure supplement 1B**). An analysis of the top 10 significantly enriched GO terms within the biological process (BP) for each B-cell subset was conducted. The results revealed varying metabolic processes enriched in each subset. For example, protein export emerged as a major BP category and the most notable KEGG pathway in plasma cells (**Figure 6—figure supplement 2E**). This outcome aligns with the understanding that

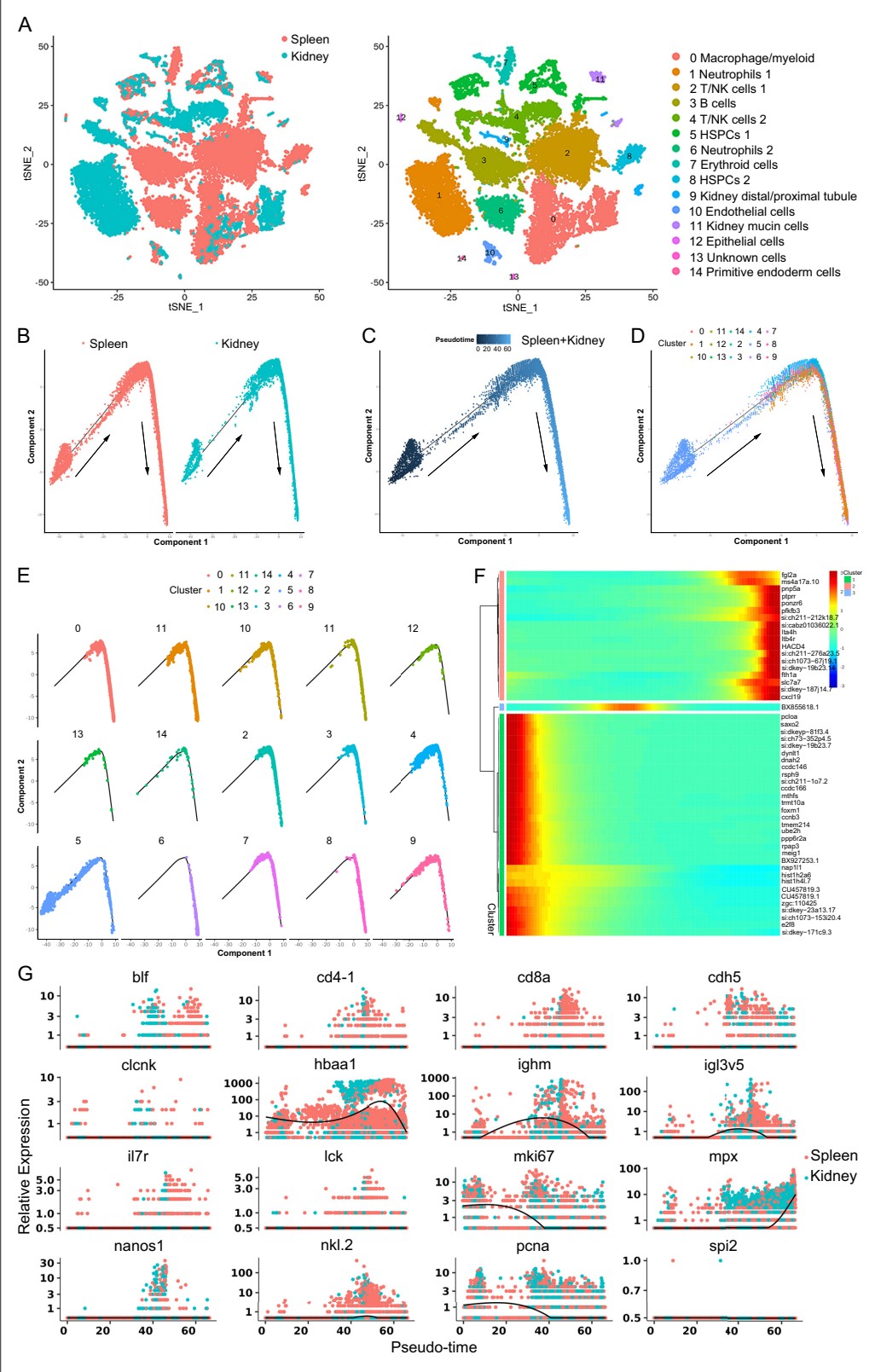

**Figure 4.** Integrated clustering and pseudotemporal analysis of zebrafish kidney and spleen hematopoietic stem/progenitor cells (HSPCs) and immune cells development. (**A**) Nonlinear t-distributed stochastic neighbor embedding (t-SNE) clustering illustrated the divergence in classification outcomes of zebrafish leukocyte populations between the kidney and spleen. (**B**) Pseudotime single-cell trajectory of zebrafish kidney marrow and

*Figure 4 continued on next page*

*Figure 4 continued*

spleen immune cells reconstructed using Monocle2. (**C**) Pseudotime single-cell trajectories of HSPCs and immune cells, with pseudotime depicted in a gradient from dark to light blue, marking its inception. (**D–E**) Distribution of subsets within the trajectory. (**F**) Pseudotemporal heatmap revealing gene expression dynamics of significant marker genes. Genes (rows) were organized into three modules, while cells (columns) followed pseudotime order. (**G**) Gene expression dynamics over pseudotime computed in Monocle2, colored by cluster assignment for 16 genes associated with immune cell development.

plasma cells are actively engaged in antibody synthesis and secretion, underscoring the importance of protein synthetic metabolism in humoral immune responses (*Zwollo et al., 2005*; *Bromage et al., 2004*). Moreover, (im)mat. B subset cells exhibited significant enrichment in antigen processing and presentation BP pathways (*Figure 6—figure supplement 2F*). This finding is congruent with prior evidence highlighting the robust antigen presentation capability of teleost fish B cells, contributing to the initiation of naïve CD4$^+$ T cell activation in adaptive humoral immunity (*Zhu et al., 2014*; *Wu et al., 2021*). Additionally, the translation process displayed enrichment across all B-cell subsets, indicative of ongoing protein synthesis during B cell development and maturation (*Figure 6—figure supplement 2A–D*). These observations provide substantive support for subset classification from the perspective of cellular metabolism. Subsequently, Monocle2 pseudotime analysis was performed on B-cell subsets using Monocle to unveil potential relationships among different subsets during B-cell development. Distinct pseudotime trajectory states of the B-cell subsets were delineated and are presented in *Figure 6D–F*. Among these subsets, pro/pre B (B2), B1-like (B8), and germinal center B-like cells (B5) occupied state 1, representing the initial differentiation phase, followed by the im/mat. B (B0 and B3), IgM$^+$IgD$^+$ B (B1), act. B/PBs (B6), and plasma (B4) cells. Notably, the identified B2, B5, and B8 clustered together in a single branch, while B0, B3, and B7 progressively differentiated into B6 and eventually into B4 (*Figure 6F*). To explore the regulation of B-cell differentiation, a pseudotime analysis was conducted on the genes significantly altered within these clusters. These genes were categorized into three modules based on their pseudotemporal expression patterns, thereby yielding potential key regulators of teleost B-cell development (*Figure 6G*, *Figure 6—figure supplement 1E*). The detailed information about the marker genes used to classify each B cell subset is provided in the *Supplementary files 1 and 5*.

## Heterogeneity of kidney macrophage/myeloid cells

We categorized kidney macrophage/myeloid cells into 10 distinct subclusters (designated as M0-M9) using a resolution of 0.3, as depicted in *Figure 7A–C*. A heatmap showcasing the top ten most significant subset-specific genes for each subcluster is provided in *Figure 7B*. M0, M1, and M7 cells were identified as M1-typical macrophages expressing marker genes such as *tnfa*, *il6*, *il1b*, *il12a*, *stat1a*, *mpeg1.2*, *mhc1zba*, *mhc2dab*, *cd83*, *cd74a*, *cd68*, *ccr7*, *selp*, and *il15ra*, suggesting their engagement in inflammatory responses (*Figure 7A–C*). Both M2 and M5 subclusters exhibited active expression of genes associated with common APC macrophages, including *mhc1zba*, *mhc2dab*, *cd83*, *cd74a* for M2, and *mpeg1.1*, *mpeg1.2*, *mhc1zba*, *mhc2dab*, *cd83*, *cd74a*, *CR318588.1*, *CR318588.2*, *CR318588.3* for M5, implying their role in antigen presentation (*Figure 7A–C*). M3 and M8 were characterized as M2-typical macrophages, demonstrated by their high expression of M2 macrophage-related genes like *cd209*, *il10*, *tgfb1a*, *tgfb1b*, *ly86* (*cd80/86*), *cd4-1*, *mpeg1.1*, *mpeg1.2*, *mhc2b*, *mhc2dab*, *cd83*, and *cd74a* (*Figure 7A–C*). M4 was classified as a macrophage-*isg15* population, expressing *mpeg1.1*, *mpeg1.2*, *mhc1zba*, *mhc2dab*, *cd83*, *cd74a*, along with interferon-inducible genes *ifi45*, *isg15*, and *ifi27.2* (*zgc:152791*), underscoring its significant role in IFN-mediated immune responses (*Figure 7A–C*). M6 cells exhibited the expression of genes related to eosinophils, including *fabp3*, *gtpbp4*, *ddx21*, *cldni*, *cd11b*, *cd44b*, *ackr3b*, indicating their resemblance to eosinophil population (*Figure 7A–C*). M9 cells represented a group of dendritic cells (DCs) extensively expressing lineage marker genes such as *zbtb46*, *scpp8*, *mmp13a.1*, *mmp9*, *mpx*, and *cpa5*. This DC group encompassed various DC populations, including classical DC (cDC), plasmacytoid DCs (pDC), and a distinct C1q$^+$DC (highly expressing *c1qa*, *c1qb*, and *c1qc*), each expressing their respective subset marker genes independently (*Figure 7A–C*; *Zhou et al., 2023*). The detailed information about the marker genes used to characterize each subcluster of macrophage/myeloid cells is provided in the *Supplementary files 1 and 6*.

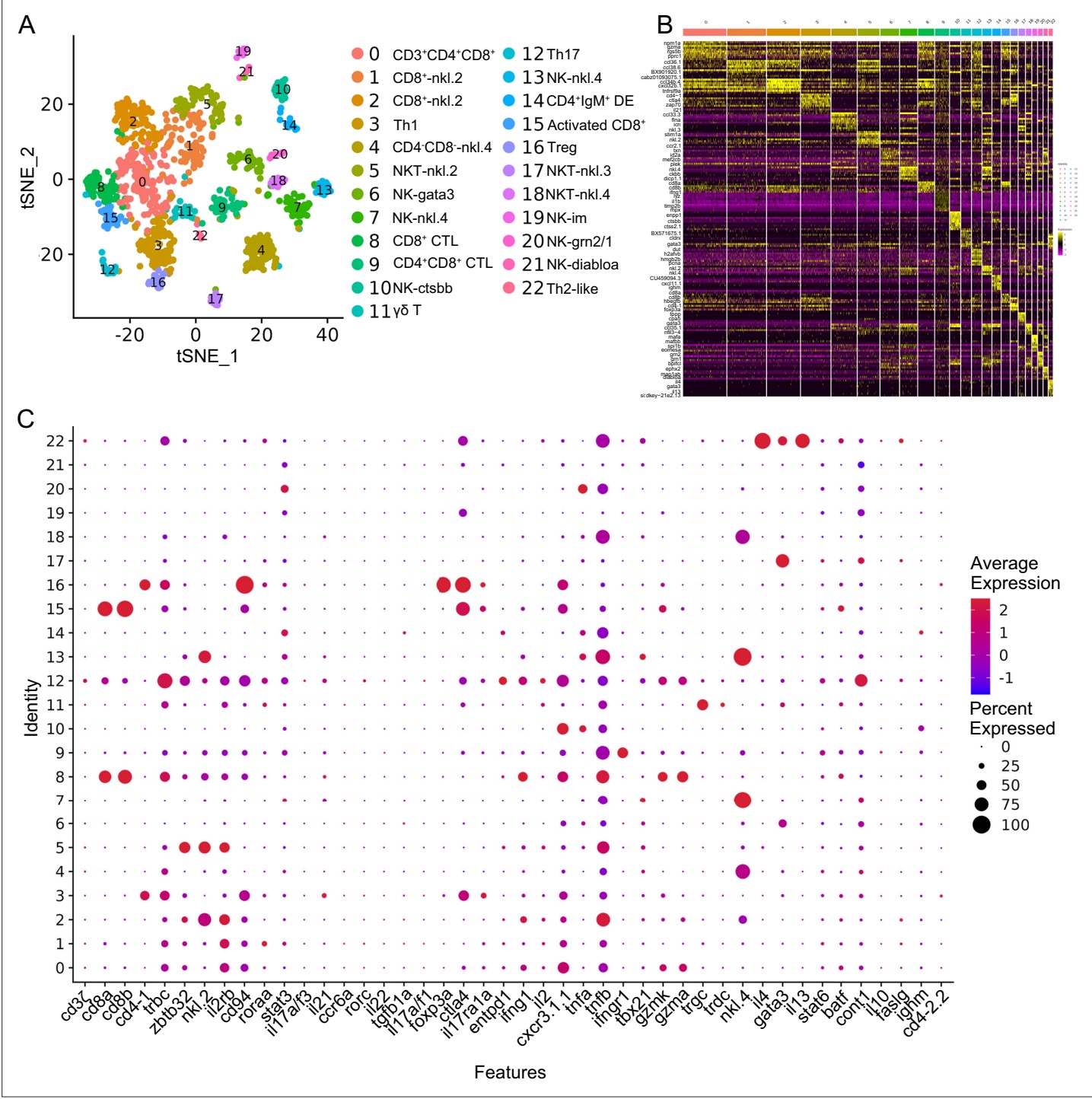

**Figure 5.** T/NK cell subtype analysis based on single-cell gene expression. (**A**) Graph-based clustering unveiled 23T/NK cell subclusters within the control group. Nonlinear t-distributed stochastic neighbor embedding (t-SNE) clustering highlighted the differing classification outcomes of T/NK cell populations in zebrafish kidney. (**B**) Heatmap showcasing marker gene expression for each cluster. Columns represent cell subtypes, while rows depict genes, color-coded according to expression levels. (**C**) Dot plots demonstrating marker gene expression levels and the respective percentages of cells per cluster expressing the gene of interest.

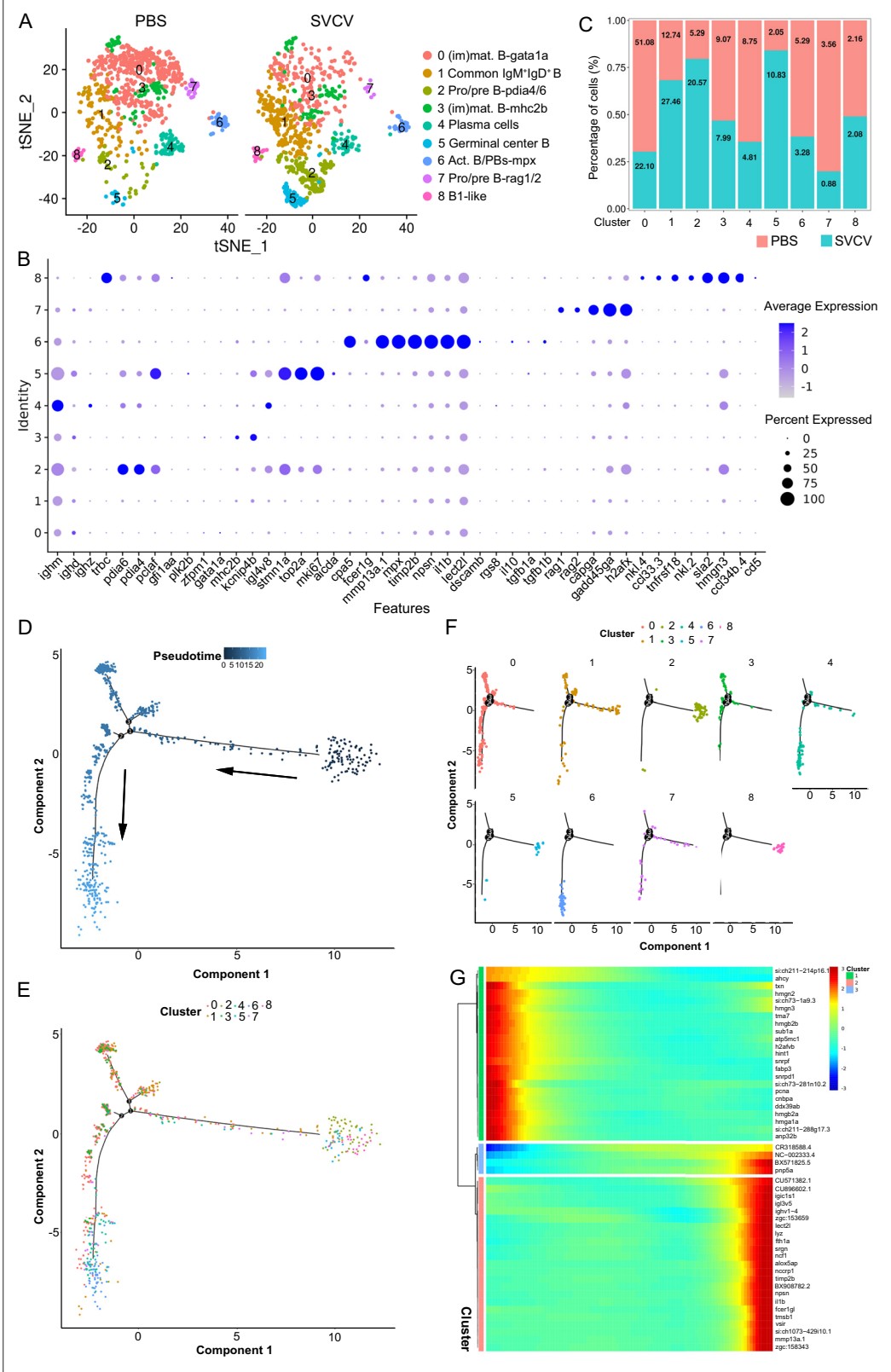

**Figure 6.** B-cell subtype and pseudotemporal analysis based on single-cell gene expression. (**A**) Graph-based clustering disclosed nine B cell subclusters in both the control group (PBS) and spring viremia of carp virus (SVCV)-infected group. (**B**) Dot plots of differentially expressed selected markers across B cell subtypes. Dot size indicates the percentage of cells within a population expressing a specific marker, while dot color indicates average

*Figure 6 continued on next page*

*Figure 6 continued*

expression. (**C**) Histogram highlighting differences in cell ratios between the control group and the SVCV-infected group. (**D**) Pseudotime single-cell trajectory of B cells reconstructed using Monocle2, pseudotime shown in a gradient from dark to light blue, marking its starting point. (**E–F**) Distribution of the nine B cell subsets within the trajectory. (**G**) Pseudotemporal heatmap revealing gene expression dynamics of significant marker genes. Genes (rows) were grouped into three modules, while cells (columns) were ordered based on pseudotime.

The online version of this article includes the following figure supplement(s) for figure 6:

**Figure supplement 1.** Identification of B cell subtype clusters and the expression patterns of marker genes.

**Figure supplement 2.** GO and Kyoto Encyclopedia of Genes and Genomes (KEGG) analysis of each cluster of B subset cells.

## Heterogeneity of kidney neutrophils

Neutrophils comprise the most prevalent subset of kidney leukocytes, constituting 45.00% within the immune-cell categories (clusters 0, 2, and 9 in *Figure 1*), and accounting for 38.15% of the total sequenced cells in both the control and SVCV-infected groups (Figure 9A and B). Subsequently, we undertook an analysis of neutrophil heterogeneity. Through iterative subclustering within the neutrophil category alone, we derived six distinct subgroups (designated as N0–N5) (*Figure 8A*). A heatmap was constructed to display the top 10 most significant DEGs within each subcluster (*Figure 8C*). Subsequent annotation of each subcluster was performed based on their unique biological functions and signaling pathways utilizing GO and KEGG analyses (*Figure 8D and E*, *Figure 8—figure supplement 1*). N0 exhibited functional enrichment in the C-type lectin receptor pathway. This subset displayed heightened expression of *tcnbb*, *marcksl1b*, *pglyrp5*, *irf1b*, *f3b*, *afap1l1b*, and *stat1a*, implying its engagement in innate antiviral immune responses (*Figure 8D*). The biological focus of N1 centered on the endocytosis pathway (*Figure 8E*). Furthermore, this population prominently expressed *scpp8*, *rab44*, *s1pr4*, *hbba1*, *crip1*, *hbaa1,* and *hbba1.1*, underscoring its pivotal role in the respiratory processes of neutrophils. N2 neutrophils were notably linked with the ribosome pathway (*Figure 8—figure supplement 1*), actively manifesting expression of *cfd*, *fabp3*, *ssr3*, *rpl22l1*, *ssr2*, *eef1g*, *rpl7*, *rplp2l*, *rpl15,* and *rpl7a* genes that are indispensable for the alternative pathway of complement activation, cytoplasmic translation, intracellular metabolism, and other facets of innate immune response. N3 subpopulation exhibited evident involvement in oxidative phosphorylation and interferon-mediated antiviral responses, as discerned from robust expression of genes such as *txn* (thioredoxin), *ifi45* and *ifi46* (interferon alpha-inducible protein 27.1 and 27.4), *si:dkey−188i13.6* (interferon alpha-inducible protein 27.3), *zgc:152791* (interferon alpha-inducible protein 27.2), *CR318588.1*, *CR318588.2*, *CR318588.3*, alongside other IFN-induced genes (*Figure 8—figure supplement 1*). N4 subcluster was typified by elevated expression of diverse immune-relevant genes, including *jdp2b*, *fosab*, and *nr4a1*, with primary functional enrichment in the MAPK signaling pathway (*Figure 8—figure supplement 1*). Finally, N5's functional enrichment revolved around the cell cycle process, evidenced by the prominent expression of *h2afx* (H2A.X variant histone), *tuba8l* (tubulin alpha 5), *top2a* (DNA topoisomerase II alpha), *mki67* (marker of proliferation Ki-67), *hist1h4l.12*, and *hmgb2b* (high mobility group box 2b), aligning closely with the architectural organization of genomic DNA (*Figure 8—figure supplement 1*). The detailed information about the marker genes used to characterize each subgroup of neutrophils is provided in the *Supplementary files 1 and 7*; *Figure 9*.

## Response of kidney HSPCs and immune cells to SVCV infection

We subsequently investigated alterations in HSPCs and immune cell categories following SVCV infection to assess the potential involvement of these cells in antiviral immunity. The results revealed considerable changes in the abundance of most categories after seven days of SVCV challenge. Specifically, macrophage/myeloid cells (cluster 1), neutrophils 2 (cluster 2), and HSPCs 2 (cluster 6) exhibited an overall increase from 12.61 to 17.30%, 3.40 to 11.70%, and 1.64 to 9.72%, respectively (*Figure 9A and B*). Conversely, neutrophils 1 (cluster 0), T/NK (cluster 5), and HSPCs 3 (cluster 8) demonstrated an overall decrease from 39.72 to 23.51%, 11.29 to 5.16%, and 6.27 to 2.08% (*Figure 9A and B*). Notably, several subset cells within the neutrophil category displayed marked changes following SVCV infection. For instance, N0, N2, and N3 cells exhibited increases from 3.45 to 41.26%, 6.75 to 17.31%, and 0.10 to 26.05%, respectively, significantly contributing to the overall rise in neutrophils 2 (cluster

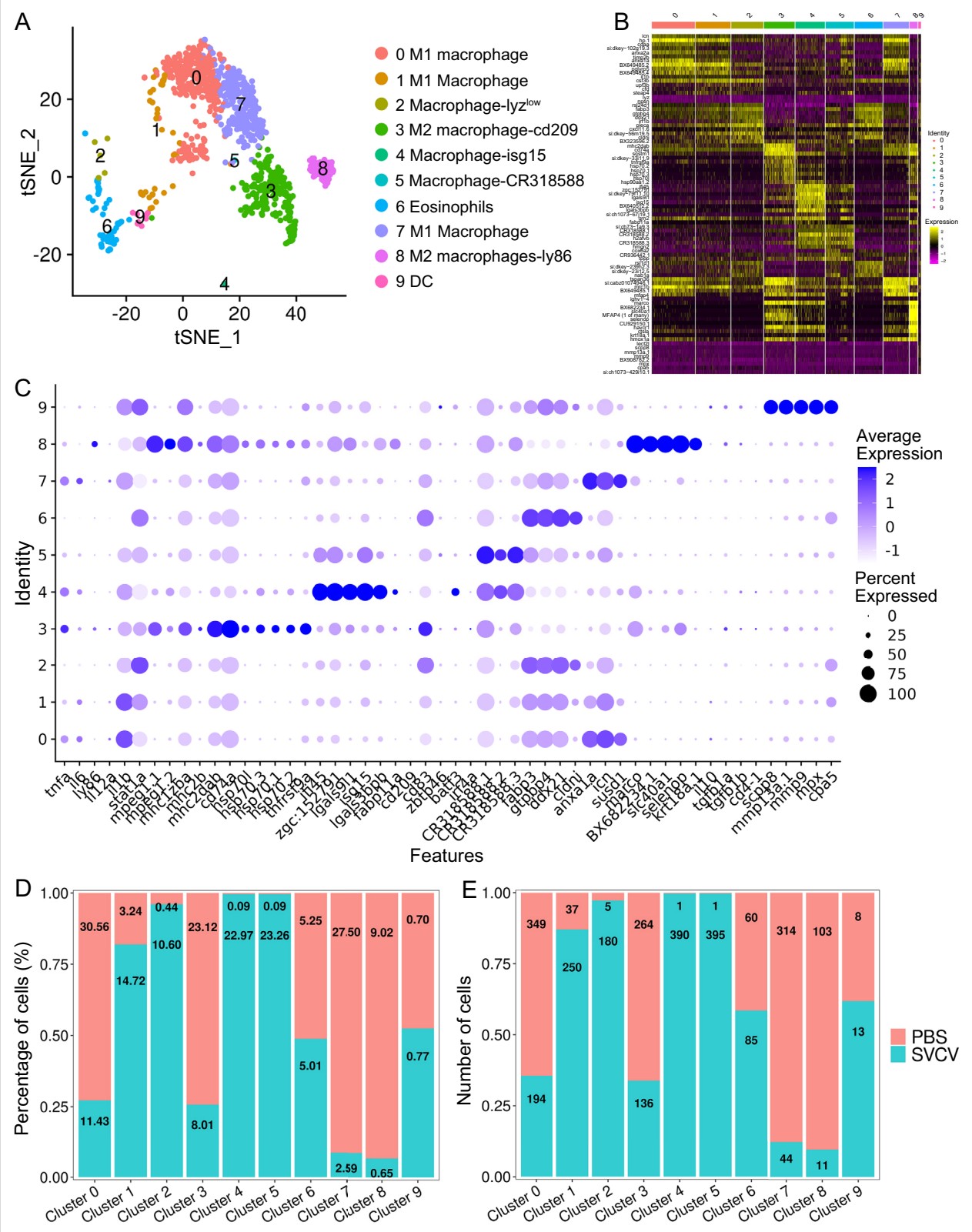

**Figure 7.** Identification of macrophage/myeloid cell subtypes. (**A**) Graph-based clustering identified 10 macrophage/myeloid cell subclusters within the control group. (**B**) Heatmap displaying marker genes for each cluster. (**C**) Dot plots illustrating marker gene expression levels and the percentages of macrophages/myeloid cells per cluster expressing specific genes. (**D**) Histogram presenting variations in cell ratios between the control group and the spring viremia of carp virus (SVCV)-infected group. (**E**) Histogram illustrating differences in cell numbers within each cluster between the control group and the SVCV-infected group.

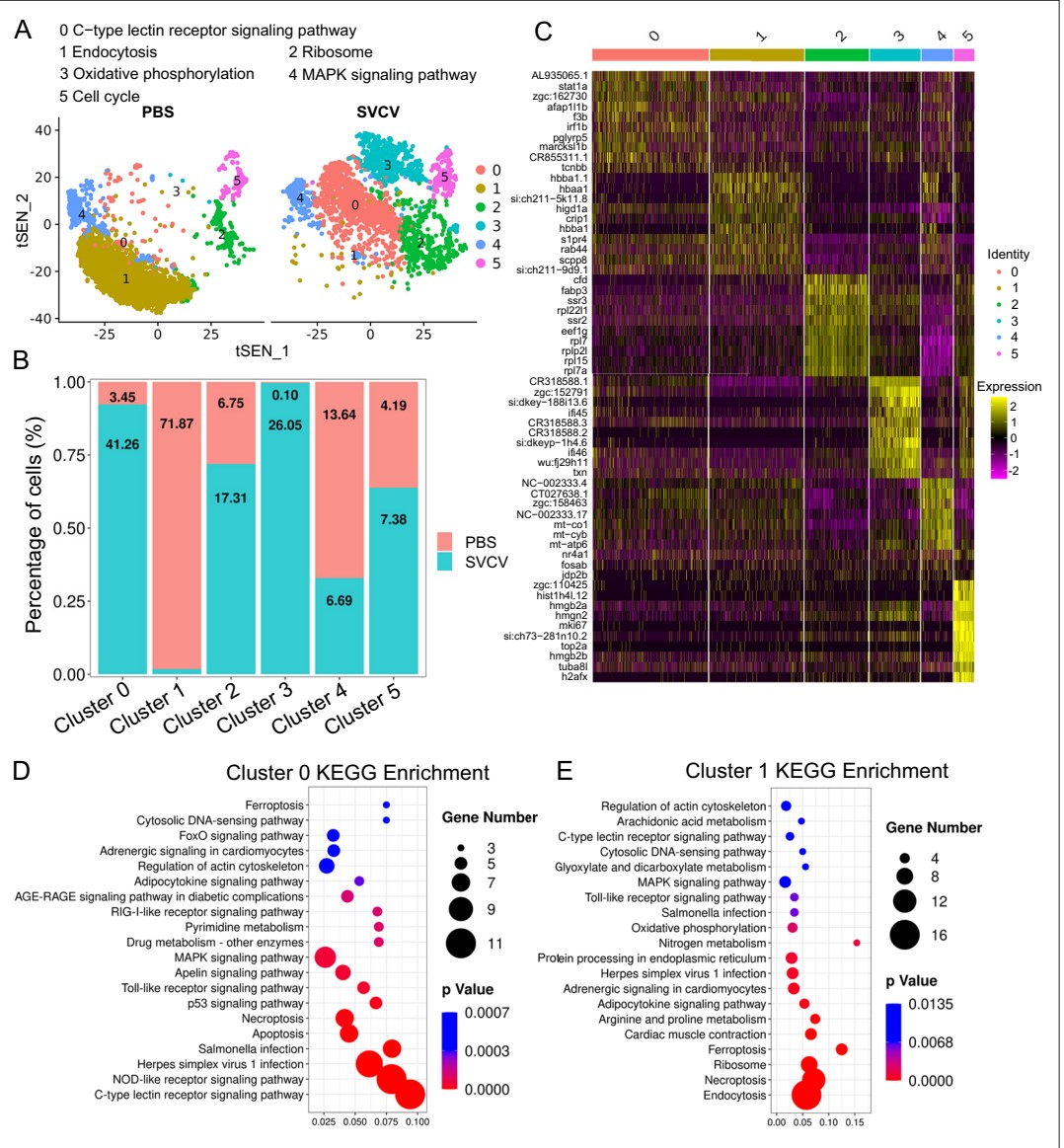

**Figure 8.** Neutrophil subtype analysis based on single-cell gene expression. (**A**) Graph-based clustering of neutrophils revealed six subclusters within both the control group (PBS) and the spring viremia of carp virus (SVCV)-infected group. Increased and decreased neutrophil subtypes were significant in the SVCV-infected group, with numeric labels indicating cluster percentages. (**B**) Histogram depicting the differing neutrophil ratios between the control group and the SVCV-infected group. (**C**) Heatmap presenting marker gene expression for each cluster. (**D**) Kyoto Encyclopedia of Genes and Genomes (KEGG) analysis outcomes for cluster 0. RichFactor signifies the ratio of differentially expressed transcripts within the pathway entry to total transcripts located in the entry, indicating enrichment. Q value represents p-value post multiple hypothesis test correction, with a lower value indicating significant enrichment. The graph sorts the top 20 pathways based on ascending Q values. (**E**) KEGG analysis outcomes for cluster 1.

The online version of this article includes the following figure supplement(s) for figure 8:

**Figure supplement 1.** Kyoto Encyclopedia of Genes and Genomes (KEGG) analysis of neutrophil subsets (**N2–N5**).

2 in *Figure 1*). By contrast, N1 and N4 cells decreased from 71.87 to 1.30% and 13.64 to 6.69% (*Figure 8A and B*). In the T/NK cell category, the abundance of NK (T6, T7, T13, T21), NKT (T5, T17), Th2-like (T22), and Th17 (T12) cells increased from 3.64 to 6.58%, 3.82 to 5.97%, 2.78 to 3.52%, 0.52 to 1.23%, 5.38 to 7.50%, 1.82 to 2.30%, 0.69 to 2.60%, and 1.65 to 3.52%, respectively (*Figure 9C*

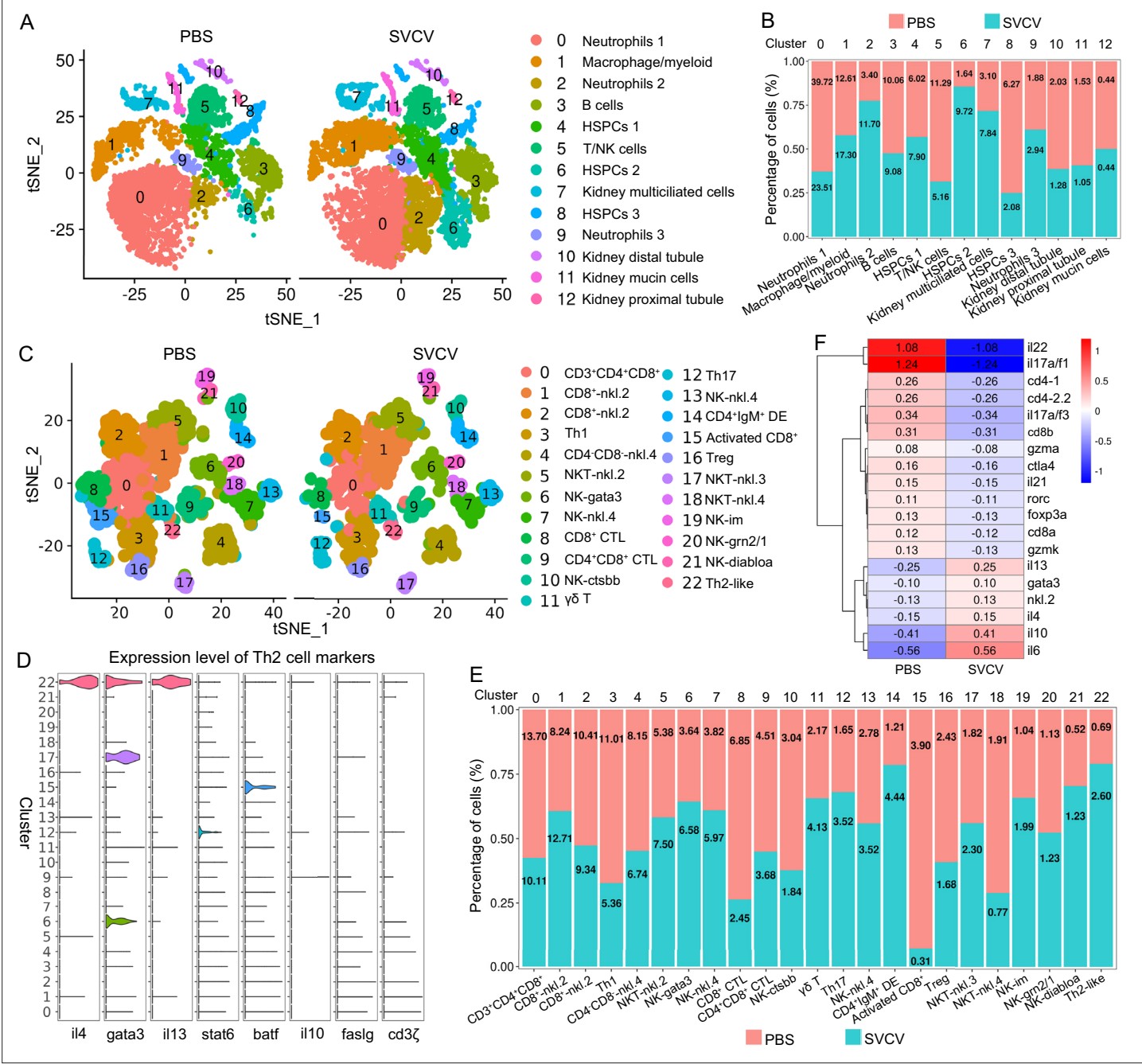

**Figure 9.** Zebrafish kidney immune-cell and T/NK cell subtype changes upon spring viremia of carp virus (SVCV) infection. (**A**) Nonlinear t-distributed stochastic neighbor embedding (t-SNE) clustering depicted variations in zebrafish kidney leukocyte classification between control (PBS) and SVCV-infected groups. (**B**) Histogram showcasing differences in cell ratios between the control group and the SVCV-infected group. (**C**) Graph-based clustering of T/NK cells demonstrated 23 subclusters within both the control group (PBS) and the SVCV-infected group. Nonlinear t-SNE clustering displayed differences in zebrafish kidney leukocyte classification between control and SVCV-infected groups. (**D**) Violin plots of Th2 cell marker genes across all clusters, displaying normalized gene expression. (**E**) Histogram showing differences in T/NK cell ratios between the control group and the SVCV-infected group. (**F**) Heatmap of significantly differentially expressed marker genes in T/NK cells between the control group and the SVCV-infected group. Gene names are indicated on the right.

*and E*). Conversely, Th1 (T3) and Treg (T16) cells decreased from 11.01 to 5.36% and 2.43 to 1.68%, accompanied by a decrease in CD8+ CTLs (T8, T9, and T15) (*Figure 9C and E*), whose activation is regulated by Th1 cells (*Huang et al., 2007*; *Ahrends et al., 2017*). Simultaneously, the expression of these cell-related genes was significantly downregulated (*Figure 9F*). In the macrophage/myeloid

category, populations such as M1, M2, M4, and M5 experienced significant increases from 3.24 to 14.72, 0.44 to 10.60%, 0.09 to 22.97%, and 0.09 to 23.26%, respectively, contributing to the overall category cell increase (*Figure 7D and E*). By contrast, M1-type (M0, M7) and M2-type (M3, M8) macrophages, decreased from 30.56 to 11.43%, 27.50 to 2.59%, 23.12 to 8.01% and 9.02 to 0.65% (*Figure 7D and E*). Among B-cell subsets, the abundance of (im)mat. B (B0) cells exhibited a significant decline from 51.08 to 22.10%, leading to a partial decrement in total B cells (*Figure 6A and C*). Correspondingly, KEGG analysis indicated that DEGs in (im)mat. B cells were primarily associated with apoptosis, ferroptosis, necroptosis, phagosome, and mitophagy, as reflected by the significant transcriptional up-regulation of related genes including *gsdmea*, *bcl2l1*, *acsl4b*, *alox5a*, *prf1.7.1*, *il1b*, *gpx4a*, *gpx4b*, *atf3*, *aco1*, *atg13*, *casp3a*, *casp3b*, *apaf1*, *casp9*, and *casp8l2* (*Figure 6—figure supplement 1C and D*), suggesting that the B0 subset experienced severe cell death upon infection. However, the proportion of common IgM⁺IgD⁺ B (B1), pro/pre B (B2), and germinal center B-like (B5) subsets increased from 12.74 to 27.46%, 5.29 to 20.57%, and 2.05 to 10.83%, respectively (*Figure 6A and C*). Within HSPC categories, cycling HSPCs (H1), HSPCs 3 (H3), and HSPCs 4 (H4) exhibited increases from 6.06 to 24.59%, 0.32 to 15.25%, and 0.32 to 16.22%, respectively. Conversely, HSPCs 1 (H0), HSPCs 2 (H2), myeloid progenitors (H6), erythrocyte progenitors (H7), lymphoid progenitors (H8), and endothelial progenitors (H9) experienced significant declines from 27.81 to 19.87%, 12.43 to 6.93%, 9.24 to 4.26%, 17.69 to 1.18%, 6.14 to 1.54%, and 11.00 to 2.10%, respectively (*Figure 2D and E*). These observations underscore the extensive response of kidney HSPCs and immune cells to SVCV infection.

## Interferon-producing cells and key virus-responsive genes

Interferons (IFNs) play pivotal roles in the defense against viral infections. In this study, we delved into the transcriptional profiles of zebrafish IFNs, specifically IFNϕ1–3 (*ifnphi1-3*), IFN-γ (*ifng1-2*), and IFNγ1R (*ifng1-1*), across various kidney HSPCs and immune cells (*Aggad et al., 2009*; *Sullivan et al., 2021*). The findings revealed varying degrees of detection for *ifnphi1*, *ifnphi2*, *ifnphi3*, *ifng1-1*, and *ifng1-2* in most HSPCs and immune cells before and after SVCV infection (*Figure 10A and B*). Pre-SVCV infection, *ifng1-1*, *ifng1-2*, and *ifnphi2* were primarily expressed in T/NK cells (cluster 5) and HSPCs 2 (cluster 6) at different basal levels, with the highest expression seen in T/NK cells (*Figure 10C*). Conversely, *ifnphi1* exhibited widespread expression in such as HSPCs 2 (cluster 6), B cells (cluster 3), HSPCs 3 (cluster 8), and macrophages/myeloid cells (cluster 1), with its peak in HSPCs 2 (cluster 6), followed by B cells (*Figure 10C*). Additionally, *ifnphi3* showed extensive expression in HSPCs 2 (cluster 6), HSPCs 3 (cluster 8), neutrophils 2 (cluster 2), B cells (cluster 3), and kidney mucin cells (cluster 11) (*Figure 10C*). Post-SVCV infection, *ifng1-1*, *ifng1-2*, *ifnphi1*, and *ifnphi3* were notably up-regulated to varying extents in most corresponding cells. Specifically, *ifnphi1* and *ifnphi3* exhibited increased expression in kidney multiciliated cells (cluster 7) and neutrophils 3 (cluster 9) upon infection (*Figure 10C*). However, the expression of *ifng1-2* in HSPCs 1 (cluster 4), HSPCs 2 (cluster 6), and HSPCs 3 (cluster 8), *ifnphi3* in kidney mucin cells (cluster 11), neutrophils 2 (cluster 2), and HSPCs 2 (cluster 6), and *ifng1-1* in HSPCs 2 (cluster 6) experienced a decline (*Figure 10C*). Overall, *ifnphi1* displayed the most significant up-regulation, while *ifng1-2* exhibited the highest expression upon challenge of SVCV, despite its downregulation trend under infection. This latter response indicated that IFN-γ actively participates in antiviral immunity alongside the previously recognized predominant role of type I IFN in antiviral defense (*Langevin et al., 2013a*; *Gan et al., 2020*). Notably, *ifnphi4* and *ifnu* were barely detectable post-SVCV infection, suggesting that IFNϕ4 and IFN-υ may play distinct functional roles in zebrafish defense against different pathogens.

We next investigated the key virus-responsive genes beyond interferons. By employing Model-based Analysis of Single-cell Transcriptomics (MAST), we identified the most significantly up-regulated genes in the DEGs of all kidney immune-cell categories. The results demonstrated a robust response of most immune cells to SVCV infection, evident from the considerable up-/down-regulation of numerous genes in these cells (*Figure 10D and E*). From the list of top-up-regulated genes, six stood out as displaying the strongest responsiveness to SVCV infection. These genes included *isg15* (ISG15 ubiquitin-like modifier), *CR318588.1*, *zgc:152791* (interferon alpha-inducible protein 27.2, *ifi27.2*), *ifi45* (interferon alpha-inducible protein 27.1, *ifi27.1*), *CR318588.2*, and *CR318588.3* (*Figure 10F–J*, *Figure 10—figure supplement 1*). Remarkably, *isg15*, *CR318588.1*, *ifi45*, and *zgc:152791* were prominent across almost every cluster (*Figure 10F*). These findings underscore the pivotal roles of these six

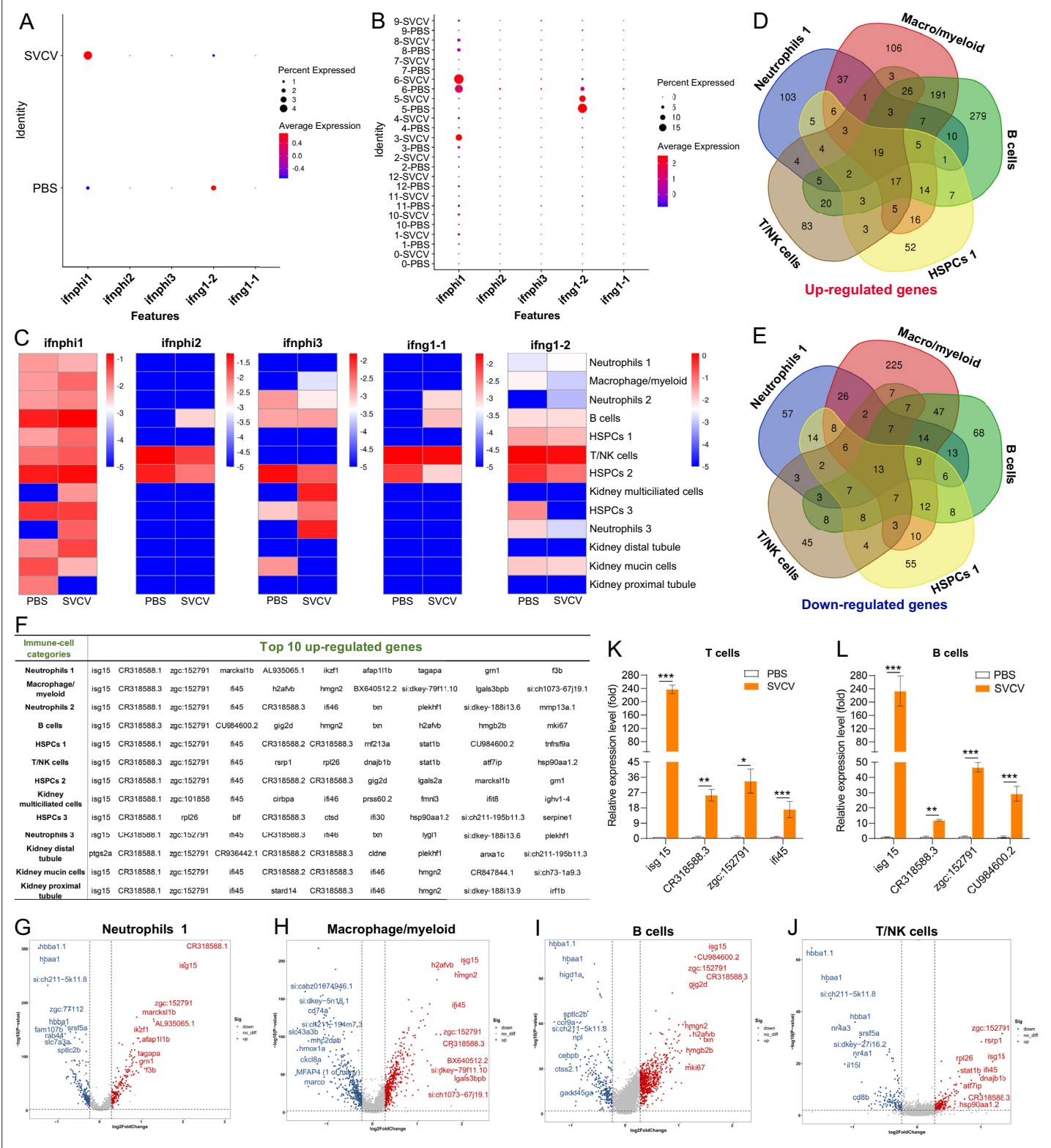

**Figure 10.** Analysis of interferons (IFN)-producing cell types and key virus-responsive genes upon spring viremia of carp virus (SVCV) infection. (**A**) Dot plots illustrating five significantly up-/down-regulated IFNs during viral infection in zebrafish kidney immune cells. (**B**) Dot plots showing five differentially expressed IFN genes in each immune cell cluster between control and SVCV-infected zebrafish. (**C**) Heatmap of five differentially expressed IFN genes in each immune cell cluster between control and SVCV-infected zebrafish. (**D–E**) Venn diagram depicting overlapping up-regulated (**D**) and down-regulated (**E**) genes among five major kidney immune-cell categories. (**F**) Top 10 up-regulated genes in each cluster of immune cells and kidney cells.

*Figure 10 continued*

**(G–J)** Volcano plot of differentially expressed genes (DEGs) in neutrophils 1 (cluster 0), macrophage/myeloid cells (cluster 1), B cells (cluster 3), and T/NK cells (cluster 5) between the control and SVCV-infected groups. **(K)** Quantitative real-time PCR (RT-qPCR) examination of four key virus-responsive genes (*isg15*, *CR318588.3*, *zgc:152791*, and *ifi45*) in T cells upon SVCV stimulation. **(L)** RT-qPCR examination of four key virus-responsive genes (*isg15*, *CR318588.3*, *zgc:152791*, and *CU984600.2*) in B cells upon SVCV stimulation. Error bars represent SD. Data was collected from a minimum of three independent experiments. *p<0.05, **p<0.01, ***p<0.001.

The online version of this article includes the following figure supplement(s) for figure 10:

**Figure supplement 1.** Volcano plots showing the differentially expressed genes (DEGs) in zebrafish kidney immune-cell and renal cell categories between control (PBS) and spring viremia of carp virus (SVCV)-infected groups (up-regulated: red; down-regulated: blue).

genes in zebrafish antiviral immunity. To validate the scRNA-seq analysis, we examined the expression levels of *isg15*, *CR318588.3*, *zgc:152791*, *ifi45*, and *CU984600.2* in kidney T-/B-cell populations using RT-qPCR. Consistently, these genes exhibited significant upregulation upon SVCV insult, confirming their responsiveness to viral infection (*Figure 10K and L*). Notably, the *isg15*, *ifi45*, and *ifi27* homologs are recognized to play crucial roles in host antiviral immunity through interaction with IFN-α/β in mammals (*Perng and Lenschow, 2018*; *Langevin et al., 2013b*). This identification of *isg15*, *ifi45*, and *ifi27* in zebrafish suggests their conserved roles in IFN-associated immunity across vertebrate evolution from fish to mammals. However, the functional roles of *CR318588.1*, *CR318588.2*, *CR318588.3*, and *zgc:152791* (*ifi27.2*, another variant of *ifi27*) in host antiviral immunity remain understudied. Consequently, further investigation is warranted to fully grasp their contributions to host immune defense against viral infections.

## Trained immune response of HSPCs to SVCV

Numerous studies have documented the phenomenon of enhanced resistance in innate immune cells such as macrophages and neutrophils upon reinfection with the same or unrelated pathogens, a phenomenon termed trained immunity (*Mitroulis et al., 2018*; *Yao et al., 2018*; *Netea et al., 2016*). Furthermore, Bacillus Calmette-Guerin (BCG) has been found to induce trained immunity in hematopoietic stem cells (HSCs) and multipotent progenitors (MPPs), promoting augmented myelopoiesis at the expense of lymphopoiesis (*de Laval et al., 2020*; *Kaufmann et al., 2018*). In this study, we investigated the potential induction of trained immunity in innate immune cells and HSPCs within the kidney upon stimulation by SVCV. To this end, zebrafish were immunized with the SVCV vaccine for 14 days and subsequently exposed to virulent SVCV for another 7 days. The outcomes demonstrated a substantial reduction in the proportion of neutrophils within cluster 0 following SVCV infection. However, this population exhibited minimal reduction in vaccinated zebrafish (referred to as "trained zebrafish") upon the second SVCV challenge (*Figure 11—figure supplement 1*). Furthermore, the abundance of HSPCs 1 within cluster 4 remained unchanged after SVCV infection, yet experienced significant expansion during the second SVCV challenge in trained zebrafish (*Figure 11—figure supplement 1*). Notably, the H0 subset exhibited the most pronounced response to trained immunity to SVCV within the HSPC categories (*Figure 11A*). Specifically, the proportion of H0 decreased from 27.81% in the PBS group to 19.87% in the SVCV group, then rapidly surged to 41.43% in the vaccinated+SVCV group. This trend was supported by the significant upregulation of numerous immune-related genes within H0 subset cells of trained zebrafish (*Figure 11B*). Thus, the shift in the H0 subset played a pivotal role in the overall heightened activity of the HSPCs 1 category within trained zebrafish. Collectively, these findings underscore an intensified immune response in kidney HSPCs of zebrafish subsequent to SVCV vaccination, suggesting the inducement of trained immunity in zebrafish kidney HSPCs for the generation of a memory response to SVCV infection. Furthermore, the neutrophils within the kidney of trained zebrafish exhibited enhanced antiviral immunity, showcasing potential innate immune memory characteristics.

## Infection of SVCV in kidney immune cells

To delineate the potential immune cells susceptible to SVCV infection, we investigated the transcription of five subgenomic RNAs/genes (sgRNAs) encoding N, P, M, G, and L proteins during the SVCV replication cycle across diverse kidney immune-cell categories, following our recently described approach (*Hu et al., 2023*). Applying an unsupervised cluster detection algorithm, we identified 10 clusters characterized by similar gene expression profiles. Importantly, the integration of viral sgRNA

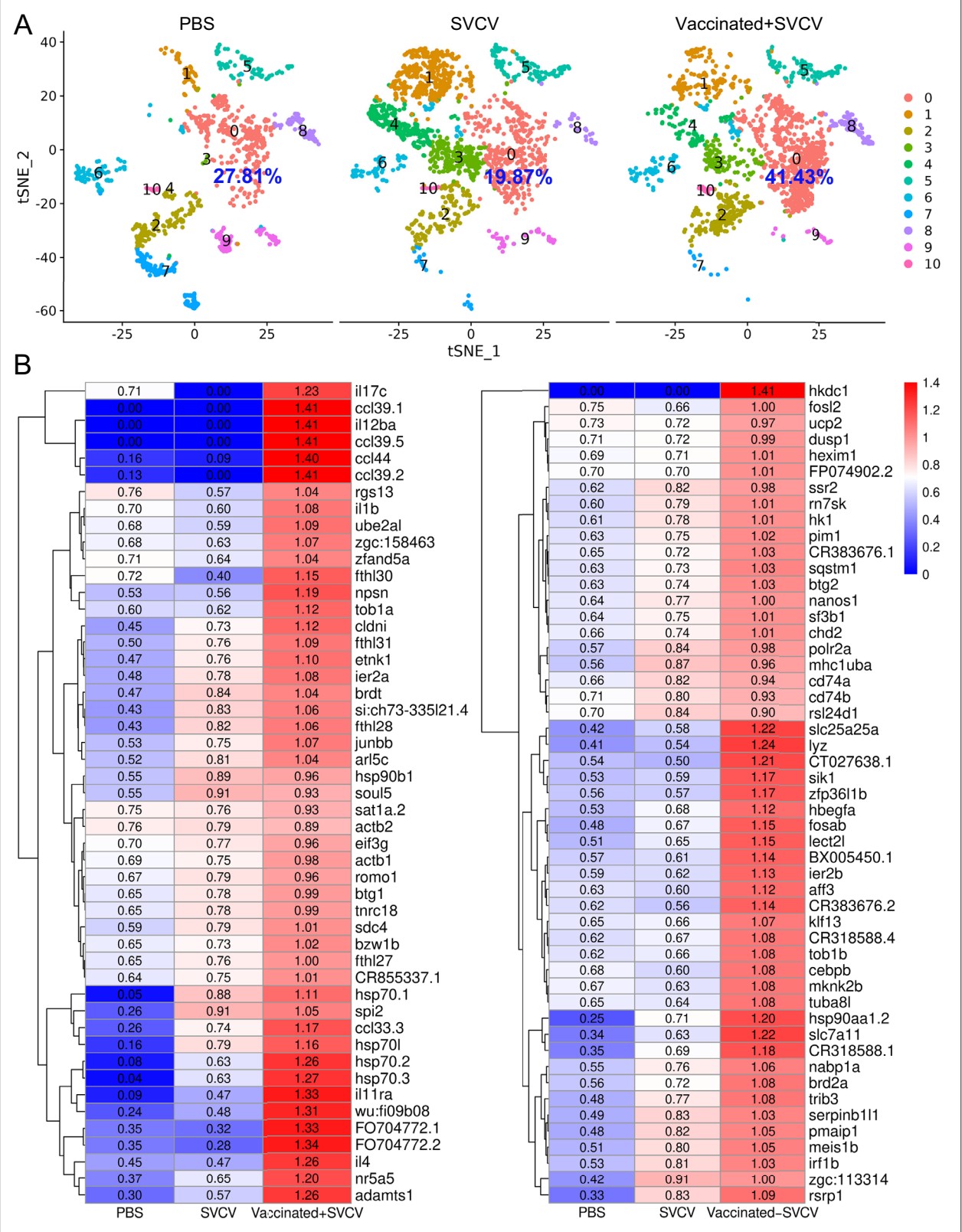

**Figure 11.** Induction of trained immunity in hematopoietic stem/progenitor cells (HSPCs) through zebrafish vaccination with inactivated spring viremia of carp virus (SVCV). (**A**) Nonlinear t-distributed stochastic neighbor embedding (t-SNE) clustering displayed the proportions of HSPCs 1 (**H0**) in the PBS-administered (control), SVCV-infected (infected), and SVCV-vaccinated plus SVCV-infected (vaccinated+SVCV) groups. (**B**) Heatmap showcasing significant immune gene expression differences in HSPCs 1 (**H0**) among the PBS-administered, SVCV-infected, and vaccinated+SVCV groups.

*Figure 11 continued on next page*

*Figure 11 continued*

The online version of this article includes the following figure supplement(s) for figure 11:

**Figure supplement 1.** Percentage and number changes in zebrafish kidney hematopoietic stem/progenitor cells (HSPCs) and immune-cell categories in response to spring viremia of carp virus (SVCV) infection.

sequences into the kidney scRNA-seq datasets revealed an overall preservation of the composition of immune-cell categories (*Figure 12A*). Subsequently, we scrutinized the transcripts of the five sgRNAs in scRNA-seq datasets from both control and SVCV-infected zebrafish. As anticipated, the five sgRNAs were present in cells from infected fish but absent in those from control fish, validating the effectiveness of our integration method (*Figure 12B*). The representation of cells transcribing N-, M-, P-, G-, and L-protein encoding sgRNAs exhibited proportional variation, with approximately 0.15% of cells carrying sgRNA for N protein, followed by M (0.12%), P (0.12%), G (0.06%), and L (0.03%) proteins, respectively (*Figure 12C*). This distribution aligns with prior observations, confirming the heightened prevalence of N protein transcripts in SVCV-infected cells, followed by those for M, P, G, and L proteins (*Riedel et al., 2020*; *Kiuchi and Roy, 1984*; *Roy, 1981*). Refining our analysis to specific cell types, we

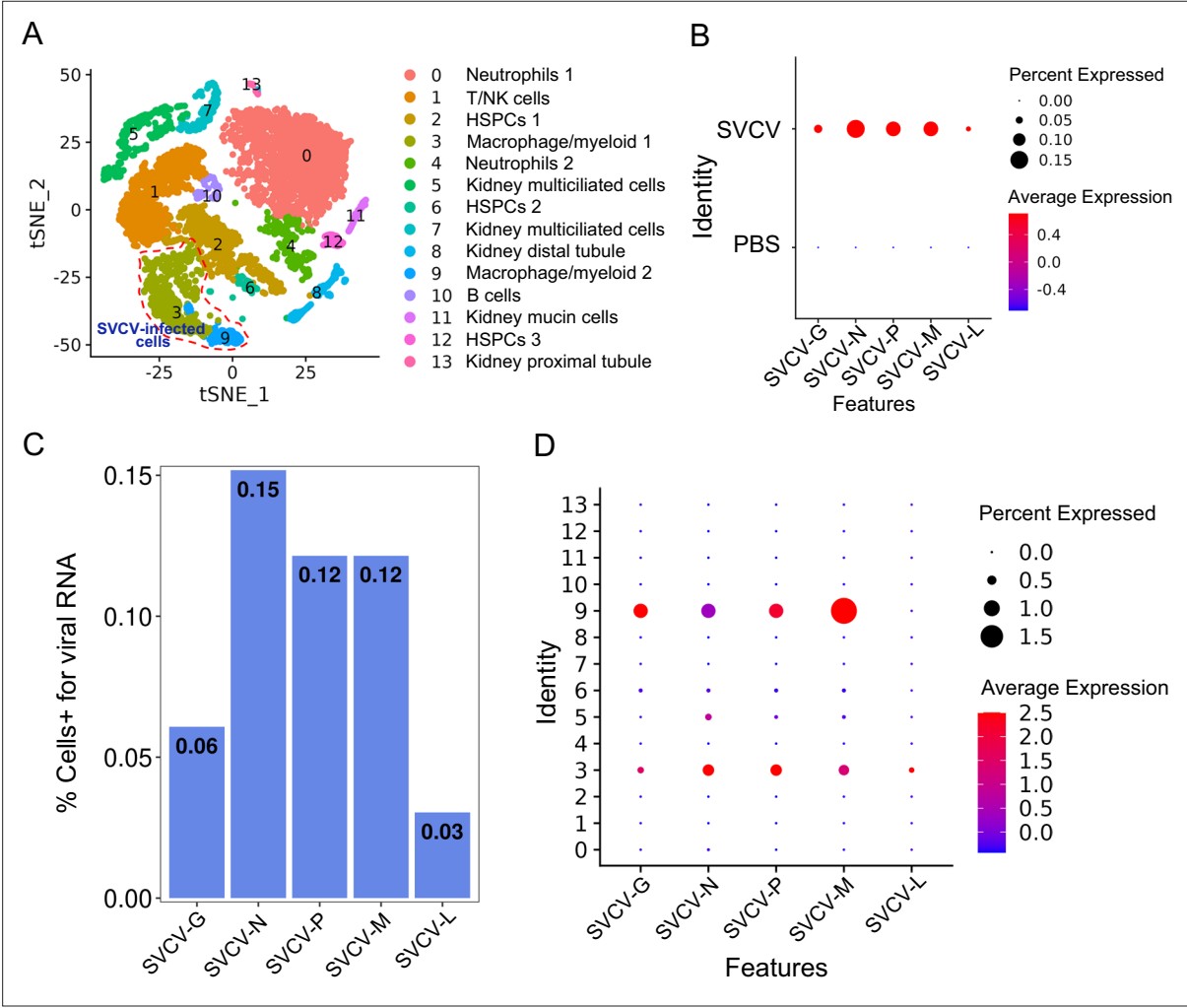

**Figure 12.** Single-cell sequencing reveals spring viremia of carp virus (SVCV)-infected cells in kidney tissue. (**A**) Nonlinear t-distributed stochastic neighbor embedding (t-SNE) clustering depicted the zebrafish kidney leukocyte classification results after integrating SVCV genes into the zebrafish genome. Principal component analysis (PCA) analysis followed data normalization, and dimension reduction was achieved via principal component analysis and graph theory-based clustering. The infected cell populations are displayed in red dashed circles. (**B**) Dot plots illustrating expression levels of five SVCV genes and the percentage of cells per cluster expressing these genes. (**C**) Percentage of single-cell RNA sequencing (scRNA-seq)-identified cells positive for reads aligning to the viral genome in SVCV-infected cases. (**D**) Dot plots showing differential expression of five SVCV genes per immune cell cluster in SVCV-infected zebrafish.

discerned that viral sgRNAs were predominantly detected in macrophage/myeloid cells within cluster 3 and 9. This observation strongly implies that macrophage/myeloid cells are the prevailing cell types supporting productive viral replication in zebrafish kidney (*Figure 12D*).

## Experimental assay for adaptive immunity occurring in kidney

The aforementioned observations underscore the immunologically responsive nature of the fish kidney, revealing innate and adaptive immune activities that extend beyond its previously recognized role in hematopoiesis. Among these activities, adaptive immunity has traditionally been attributed to secondary lymphoid organs such as the spleen and lymph nodes (*Cooper and Alder, 2006*; *Boehm and Bleul, 2007*). This study offers experimental confirmation for the occurrence of adaptive immunity in the zebrafish kidney following viral stimulation. To this end, zebrafish were exposed to inactivated SVCV ($10^6$ TCID50/fish) for 7 days, and the activation of kidney T and B cells was assessed by quantifying their heightened proliferation and increased expression of CD154 and CD40 costimulatory molecules through flow cytometry and RT-qPCR. As anticipated, CD4$^+$ T, CD8$^+$ T, and IgM$^+$ B cells exhibited significant proliferation in the kidneys of zebrafish stimulated with SVCV compared to control fish that received mock PBS (*Figure 13A–D*). In tandem, the transcriptional levels of CD154 and CD40 genes displayed marked upregulation, concurrent with the proliferation of CD4$^+$ T, CD8$^+$ T, and IgM$^+$ B cells (*Figure 13E and F*). Importantly, parallel activation of CD4$^+$ T, CD8$^+$ T, and IgM$^+$ cells was also discerned in the spleens of SVCV-stimulated zebrafish (*Figure 13G*). These results suggested that the zebrafish kidney assumes a central role in orchestrating adaptive immune responses to viral infection, a role that may be comparable to that of the spleen, a well-known secondary lymphoid organ crucial for initiating and advancing adaptive immunity. To substantiate this perspective, we proceeded to investigate the occurrence of SHM, a hallmark event driving antibody affinity maturation in adaptive humoral immunity, within kidney B cells upon antigenic stimulation. We employed RT-qPCR to examine the expression of the zebrafish AID gene (*aicda*), a pivotal initiator of SHM. As anticipated, AID transcript presence was evident in the zebrafish kidney, and its expression significantly increased in response to SVCV stimulation (*Figure 13H*). Correspondingly, SHM was identified in kidney B cells through PCR-based high-throughput sequencing. Notably, the frequency of mutagenesis within V regions (V4-9) of IgM genes demonstrated a substantial antigen-dependent increase (*Figure 13I*). These findings offer novel insights into the dual-functionality of the zebrafish kidney, encompassing both hematopoietic and adaptive immunological responsiveness. This suggests that the fish kidney operates as a unique organ amalgamating the roles typically associated with primary and secondary lymphoid tissues.

## Discussion

The immune system constitutes a complex evolutionary entity marked by significant heterogeneity and structural-functional diversity across various taxa. The origin and phylogenetic evolution of the vertebrate immune system captivates the attention of comparative immunologists, particularly concerning teleost fish due to their pivotal phylogenetic standing. In human and other mammalian organisms, the immune system encompasses primary and secondary lymphoid organs (*Neely and Flajnik, 2016*; *Boehm and Bleul, 2007*), the former including bone marrow and thymus, facilitating hematopoiesis, lymphocyte maturation, and functional development (*Boehm and Bleul, 2007*; *Chinn et al., 2023*). Secondary lymphoid organs, including lymph nodes, spleen, and sections of mucosal immune systems, trigger the activation of naïve lymphocytes by antigens, thereby initiating and advancing adaptive immune responses (*Neely and Flajnik, 2016*; *Malhotra et al., 2013*; *Randall et al., 2008*). In contrast to mammals, teleost fish lack bone marrow and lymph nodes. Instead, the anterior section of the fish kidney (referred to as the head kidney) emerges as an operational analog of mammalian bone marrow in adult fish. Accordingly, the fish kidney functions as the primary hematopoietic site, overseeing the generation of circulating blood cell lineages and B cell maturation (*Stachura et al., 2009*; *Diep et al., 2011*; *Page et al., 2013*). Furthermore, the fish kidney is also thought to be an immunologically responsive organ capable of eliciting immune responses against a spectrum of pathogens, including bacteria, viruses, and parasites (*Zwollo et al., 2005*; *Bromage et al., 2004*). These findings establish the kidney's pivotal role in fish immunity, although a comprehensive understanding of the

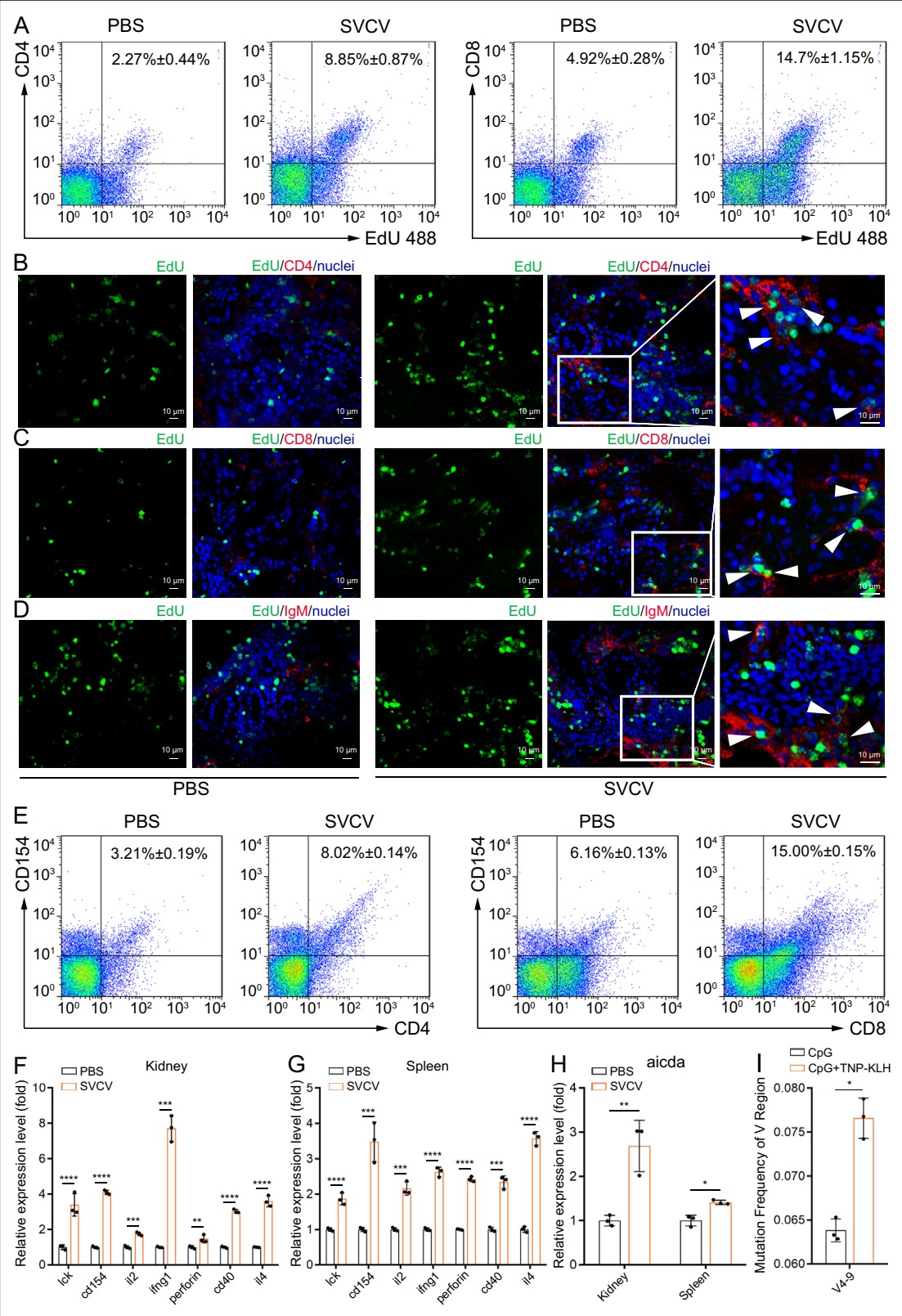

**Figure 13.** Examination of zebrafish kidney acting as a secondary immune organ involved in the antiviral immune response. (**A**) In vivo proliferation assessment of CD4⁺ and CD8⁺ T cells in the kidney of zebrafish stimulated with inactivated spring viremia of carp virus (SVCV) ($10^6$ TCID$_{50}$), determined by CD4⁺EdU⁺ and CD8⁺EdU⁺ T cell percentages via flow cytometry (FCM) analysis. (**B–D**) Analysis of proliferative CD4⁺ T, CD8⁺ T, and IgM⁺ B cells by immunofluorescence staining. Kidney cryosections were stained for EdU (green), CD4 (red, **B**), CD8 (red, **C**), and IgM (red, **D**), and

*Figure 13 continued on next page*

*Figure 13 continued*

nuclei (blue) detection. Fluorescence images were captured using a Laser scanning confocal microscope (FV3000, 60x oil). White arrowheads point to cells double-stained for EdU with CD4, CD8, or IgM. Scale bars, 10 μm. (**E**) In vivo activation examination of CD4$^+$ and CD8$^+$ T cells after SVCV stimulation, determined by CD4$^+$CD154$^+$ and CD8$^+$CD154$^+$ T cell percentages via FCM analysis. (**F–G**) Quantitative real-time PCR (RT-qPCR) analysis of transcriptional expression of T/B cell activation-related genes in kidney and spleen under SVCV stimulation. (**H**) RT-qPCR analysis of B cell somatic hypermutation (SHM)-related gene AID (*aicda*) transcriptional expression in kidney and spleen under SVCV stimulation. (**I**) SHM analysis in IgM antibody V genes through multiple PCR-based high-throughput sequencing. Experiments were independently repeated at least three times (*p<0.05, **p<0.01, ***p<0.001, ****p<0.0001).

cellular components and their attributes within the hematopoietic process and immune responses in the fish kidney remains elusive.

To address these gaps, we conducted a single-cell RNA sequencing (scRNA-seq) analysis of kidney leukocytes in zebrafish, unraveling the spectrum of hematopoietic and immune-associated cell types in adult fish. We identified 13 distinct cell categories, spanning three groups of HSPCs, six cohorts of innate and adaptive immune-related cells, and four renal cell types, which emerged as byproducts of cellular preparation. Further subdivisions of the three HSPC groups led to the identification of 11 potential subpopulations, each marked by unique gene expression profiles. Notable subsets, such as Nanos1$^+$Spi2$^+$ HSPCs and cycling HSPCs, bore resemblances to previously described counterparts (*Rubin et al., 2022*). Moreover, six subsets exhibited the characteristics of progenitor cells, demonstrating both marker gene expression and the capacity to differentiate into diverse blood cell lineages encompassing NK cells, myeloid cells, lymphoid cells, erythrocytes, and thrombocytes. Yet, the properties of certain HSPC subsets, like H2, H3, and H4, remain undefined. Such subsets exhibit distinct gene expression profiles, which could potentially serve as candidate signatures for future subset classification. Correspondingly, numerous committed or terminal cell lineages, intricately associated with these HSPCs, were detected in the kidney. These lineages encompass a range of innate and adaptive immune-related cells at various developmental stages, prominently featuring neutrophils, macrophages, dendritic cells, NK cells, and various T and B cell subsets. Moreover, early developmental cells such as double negative (DN) T cells, double positive (DP) T cells, pro/pre B cells, and immature/mature B cells were found in the kidney. These cells are major committed immature or partially mature T and B lymphocytes at the early stage of hematopoiesis, among them, the DN and DP T cells will enter the thymus, where they complete their maturation and maturing selection (*Rubin et al., 2022*), while pro/pre B and im/mat B cells may continue to mature or partially mature in the kidney. These insights underscore the presence of diverse HSPCs fostering various blood cell lineages and reinforce the concept that the fish kidney functions as a primary lymphoid organ, mirroring the role of mammalian bone marrow (*Supplementary file 8*; *Catton, 1951*; *Zapata, 1979*).

Comparatively, human hematopoietic stem cells (HSCs) are characterized by surface markers (such as CD34 and c-kit), coupled with the absence of lineage-specific markers typical of mature cells (*Yang et al., 2005*; *Shin et al., 2014*). HSCs yield common myeloid-lymphoid progenitors (CMLPs) and common myeloid-megakaryocyte-erythroid progenitors (CMMEPs). While CMLPs give rise to committed T and B cell precursors, ILCs, and certain myeloid cells, CMMEPs produce precursors for erythroid, megakaryocytic, monocytic, and granulocytic lineages. These precursors subsequently develop into mature erythrocytes, platelets, monocytes, and granulocytes, as well as dendritic cells and mast cells (*Woolthuis and Park, 2016*; *Kondo, 2010*). Notably, zebrafish exhibited detectable elementary progenitors responsible for generating principal blood cell lineages, suggesting the presence of conserved cellular components and developmental pathways underlying adult hematopoiesis across species. However, unresolved matters and fish-specific events within adult fish hematopoiesis persist, such as distinguishing HSCs from multiple HSPCs. Additionally, the existence of pluripotent common progenitors akin to CMLPs and CMMEPs, as well as committed precursors for lymphocytic, erythroid, megakaryocytic, monocytic, and granulocytic lineages, warrant clarification. Although pseudotime trajectory and RNA velocity analyses have shed preliminary light on relationships among partial HSPCs, the precise phenotypic features, developmental trajectories, and correlations of these HSPCs necessitate thorough elucidation. Unique HSPCs with unknown properties, like H0, H2, H3, and H4, accentuate the high heterogeneity within HSPCs in adult fish. Given the considerable number of fish-specific lineages, probing the relationship between these uncharacterized HSPCs and fish-specific lineages remains a promising avenue of exploration.

The extensive presence of immune cells associated with the kidney signifies the central role of this organ in fish immunity, extending beyond its hematopoietic function. Indeed, the kidney exhibited robust responsiveness to SVCV infection, evidenced by significant changes in the abundance of diverse immune cells, including both upregulated and downregulated subsets. Prominent upregulated populations consisted of neutrophils (N0, N2, N3), NK (T6, T7, T13, T21), NKT (T5, T17), Th17 (T12), Th2-like (T22), common IgM$^+$IgD$^+$ B (B1), pro/pre B (B2), germinal center B-like (B5) cells, and macrophages (M1, M2, M4, M5), indicating their engagement in host defense against SVCV. Conversely, downregulated populations encompassed subsets of neutrophils (N1, N4), CD8$^+$ T (T8, T9, T15), Th1 (T3), Treg (T16), im/mat. B (B0), and M1-/M2-macrophages (M0, M3, M7, M8), implying their impairment during SVCV infection. For example, the impairment of im/mat. B (B0) cells may be caused by undergoing severe cell death under infection in this case, as determined by their high expression of death-associated genes related to apoptosis, ferroptosis, and necroptosis. The reduction in this antibody-producing subset will attenuate the humoral adaptive immune protection of fish from SVCV infection. Notably, we found that the abundance of three HSPC subsets (H1, H3, and H4) were markedly upregulated by 24.09-, 255.22-, 258.44-folds, respectively. By contrast, six other HSPC subsets (H0, H2, H6, H7, H8, and H9) were downregulated, potentially hampering the differentiation of downstream lineage cells and attenuating antiviral immunity. Encouragingly, two HSPC subsets demonstrated the induction of trained immunity upon stimulation with SVCV vaccine, subsequently challenged by virulent SVCV. These findings not only underscore the role of HSPCs in viral responses but also signify a novel dimension of HSPC involvement in viral infections. Previous studies have highlighted the profound impact of infections on host immune homeostasis in various viral disease models (*Sumbria et al., 2020*; *Rouse and Sehrawat, 2010*; *Perelson and Ribeiro, 2018*). The consequences range from excessive inflammation due to hyperactive immune responses to immune deficiency resulting from ineffective responses, the latter of which potentially favors immune escape from viruses (*Rouse and Sehrawat, 2010*). In the current study, multiple HSPCs and immune cell subset disturbances were observed under SVCV infection, suggesting a potential role in disease pathogenesis (*Spellberg and Edwards, 2001*). However, the exact mechanisms of the disturbance of HSPCs and immune cells under infection remain to be studied, which will help clarify the pathogenesis of viremia during SVCV infection from the perspective of immunopathology.

The broader overview of kidney immune cell shifts in response to SVCV challenge reveals extensive innate and adaptive immune responses within the kidney during viral infections. This robust immune responsiveness aligns with the concept of the fish kidney as a potent immune organ (*García-Valtanen et al., 2017*). Particularly noteworthy, our investigation revealed a diverse range of adaptive immune-associated cells inhabiting the zebrafish kidney. These cells encompass various APCs, along with naïve T and B cells, plasmablasts, and plasma cells—key cell types crucial for initiating, activating, and executing adaptive immune responses. The profiles of adaptive immune cells within the kidney, alongside their responses, closely resemble those observed in the spleen—an evolutionarily conserved secondary lymphoid organ spanning vertebrates from fish to mammals. Consequently, our findings strongly imply that the zebrafish kidney might function as a secondary lymphoid organ, fostering the generation of adaptive immunity through interactions among APCs and antigen-specific T and B cells. To empirically substantiate this hypothesis, we examined the occurrence of antigen-stimulated adaptive immune responses in the kidney. As anticipated, the CD4$^+$ T, CD8$^+$ T, and IgM$^+$ B cells exhibited pronounced activation in the zebrafish kidney under SVCV challenge. This activation was underscored by significant proliferation in CD4$^+$ T, CD8$^+$ T, and IgM$^+$ B cell populations, followed by upregulated transcriptional expression of CD154 and CD40 genes in these cells. Furthermore, we also observed the occurrence of antigen-induced expression of AID and SHM in the kidney IgM$^+$ B cells, two hallmark events for the affinity maturation of antibodies in adaptive humoral immunity. These results collectively establish the zebrafish kidney as a distinct entity encompassing attributes of both primary and secondary lymphoid organs, thus offering fresh insights into the dual functional roles of the fish kidney, spanning hematopoiesis and adaptive immunity. While peripheral lymph nodes are absent, the spleen has historically stood as the predominant secondary lymphoid organ in fish. However, our current study introduces a novel member to the array of secondary lymphoid organs. The intricate interplay between the roles of the kidney and spleen in hematopoiesis and immune responses within fish warrants further elucidation. Moreover, unraveling the structural foundations accommodating diverse HSPCs and immune cell types and precisely orchestrating

hematopoiesis and immune responses within the kidney and spleen emerges as a compelling area of inquiry. Addressing these queries necessitates the utilization of advanced techniques, such as single-cell spatiotemporal transcriptomic analysis at the level of cross-organ single-cell transcriptome profiling.

In summary, our study presents a comprehensive panorama of HSPCs and immune cell types within the zebrafish kidney under steady-state conditions, viral emergencies, and vaccine training. Our identification of a multitude of HSPCs, along with developmentally-linked progenitors and immune cell types, and their modifications under emergency and training scenarios, underscores the pivotal role of the kidney in fish defense against viral infections. Moreover, the revelation of trained immunity establishment within HSPCs holds significant implications for comprehending the novel pathways and mechanisms governing trained immunity formation. The paradigm shift—where the kidney, traditionally viewed as a primary lymphoid organ facilitating hematopoiesis—emerges as a secondary lymphoid organ, marks a pivotal transformation in understanding fish biology. Our study offers insights into the intricate and heterogeneous nature of the fish immune system, shedding light on the multifunctionality of the fish kidney, encompassing hematopoiesis, immune defense, waste excretion, and osmoregulation—mainstays of kidney function across diverse vertebrates.

## Materials and methods
### Experimental fish
Wild-type AB zebrafish (*Danio rerio*) of both sexes were obtained from China Zebrafish Resource Center (Wuhan, China), 3–12 months of age with body weights ranging from 0.5 to 1.0 g and lengths of 3–4 cm, were reared under standard laboratory conditions. The fish were raised in recirculating water at a temperature of 28 °C and exposed to a 12 hr light/12 hr dark cycle, as previously described (*Wan et al., 2016*; *Li et al., 2020*). Prior to their use in experiments, all fish were acclimated for a minimum of 2 weeks to ensure their overall health. Only zebrafish exhibiting robust general appearance and activity were selected for experimentation. All animal experiments were conducted in accordance with the guiding principles for the care and use of laboratory animals and were approved by a local ethics committee.

### Virus
The spring viremia of carp virus (SVCV), provided as a gift by Prof. Yibing Zhang from the Institute of Hydrobiology, Chinese Academy of Sciences, was propagated in epithelioma papulosum cyprini (EPC) cells at 28 °C. The virus was subsequently titrated in 96-well plates until full cytopathic effect (CPE) was observed. Purification of SVCV from EPC cell culture was accomplished through sucrose gradient centrifugation at 100,000 g for 2 hr, as previously described (*Chen et al., 2008*). The purified virus was suspended in sterile PBS and stored at –80 °C until utilization. The viral titer was quantified as $1.58 \times 10^7$ TCID$_{50}$/ml, employing the 50% tissue culture infective dose (TCID50) assay on EPC cells via the Reed-Müench method (*Reed and Muench, 1938*). For the purpose of vaccination, SVCV was inactivated with a final concentration of 0.4% (v/v) formaldehyde, followed by the protocol as described (*Dong et al., 2013*).

### SVCV challenge
Adult zebrafish were randomly divided into three groups, i.e., PBS-administered group (control), SVCV-infected group (infected), and inactivated SVCV-vaccinated plus virulent SVCV-infected group (vaccinated +infected). Each zebrafish in these groups received an intraperitoneal injection of 10 μl of phosphate-buffered saline (PBS, pH = 7.0), 10 μl of virulent SVCV ($10^3$ TCID$_{50}$/fish), or 10 μl of inactivated SVCV followed by 10 μl of virulent SVCV ($10^3$ TCID$_{50}$/fish). In the context of vaccination, zebrafish were immunized with 10 μl of inactivated SVCV vaccine ($10^5$ TCID$_{50}$/fish) for two weeks, succeeded by infection with virulent SVCV ($10^3$ TCID$_{50}$/fish) for an additional week. Subsequently, kidney samples were collected at 7 days post-infection (dpi) for the control and SVCV-infected groups and at 21 dpi for the vaccinated +infected group. Equivalent quantities of tissue were pooled from each group, and these samples underwent subsequent leukocyte isolation.

## Isolation of leukocytes

Leukocytes were obtained from kidney samples utilizing Ficoll-Hypaque density-gradient centrifugation, as previously described (*Wan et al., 2016*; *Zhu et al., 2014*). After 7 days of SVCV challenge, zebrafish were euthanized with MS222. Kidney tissues were meticulously excised and passed through a 40 μm stainless nylon mesh (Greiner Bio-One GmbH, Germany). The resulting cell suspension was suspended in Leibovitz's L-15 Medium (L-15, Gibco), supplemented with penicillin (100 U/ml, Sigma-Aldrich), streptomycin (100 μg/ml, Sigma-Aldrich), and heparin sodium (10 U/ml, Sigma-Aldrich). This suspension was then introduced into a Ficoll-Hypaque (1.080 g/mL) density-gradient centrifugation, and subsequently centrifuged at 1200 g for 25 min. The cellular interface layer was cautiously aspirated, followed by a wash with ice-cold PBS at 400 g for 10 min. The viability and quantity of cells were assessed using 0.4% trypan blue (Sigma, St. Louis, MO, USA), confirming the viability of over 95% of the cells. Enumeration of cells for each group was executed using a cell counting plate.

## Single-cell RNA sequencing

The library synthesis and RNA-sequencing were completed by the LC-Bio Technology Co., Ltd (Hangzhou, China). In brief, the isolated leucocytes from zebrafish kidney were applied to Red Blood Cell Lysis Buffer and the magnetic bead separation method was used to remove red blood cells and dead cells. After cell counting, an appropriate volume for capturing 10,000 cells was calculated. These cells were mixed into a sample and adjusted to 700~1200 cell/μl. Libraries were constructed using the Chromium Controller and Chromium Single Cell 3' Library & Gel Bead Kit v2 (10×Genomics) as per the manufacturer's protocol. Cellular suspension was added to the master mix containing nuclease-free water, RT Reagent Mix, RT Primer, Additive A and RT Enzyme Mix. Master mix with cells was transferred to the wells in the row labeled 1 on the Chromium Single Cell A Chip (10×Genomics). Single Cell 3' Gel Beads and Partitioning Oil were placed in specific rows. The chip was loaded onto the Chromium Controller to create single-cell GEMs. GEM-RT was conducted in a C1000 Touch Thermal cycler (Bio-Rad), followed by cleanup with DynaBeads MyOne Silane Beads (Thermo Fisher Scientific). Subsequent cDNA amplification and library construction adhered to standard protocols. The library was multiplexed and sequenced on an Illumina NextSeq-500 lane using a high-output (400 m) kit (*Hernández et al., 2018*). Quality control of 10x Genomics single-cell RNA-seq was executed (*Figure 1—figure supplement 1*). For mapping, sequences from the 10×Genomics single-cell RNA-seq platform were demultiplexed and mapped to the zebrafish genome (GRCz11) through the Cell Ranger package (10x Genomics). The DoubletFinder R package identified and filtered doublets (multiplets) (*McGinnis et al., 2019*). The raw digital gene expression matrix (UMI counts per gene per cell) from Cell Ranger analyses was filtered and normalized using the R package Seurat (Version 3.1.1) (*Butler et al., 2018*). Cells with fewer than 500 unique genes, over 50,000 UMI counts, or more than 10% mitochondrial reads were removed. Undetected genes in any cell were excluded. Post-filtration, cell counts were 9004 in the PBS-administered group (control), 9890 in the SVCV-infected group (infected), and 8487 in the SVCV-vaccinated plus SVCV-infected group (vaccinated+infected).

## Downstream analysis of 10x Genomics data

After log normalizing the data, a principal component analysis (PCA) was performed to reduce dimension. The identification of significant clusters was performed using the Find Clusters algorithm in the Seurat package, which uses a shared nearest-neighbor modularity optimization-based clustering algorithm. Marker genes for each significant cluster were found using the Seurat function FindAllMarkers. Cell types were determined using a combination of marker genes identified from the literature and gene ontology for cell types. Expression of selected genes was plotted with the Seurat function FeaturePlot and VlnPlot. Hierarchical clustering and heat map generation were performed for single cells based on log-normalized (with scale factor 10,000 and pseudocount 1) expression values of marker genes obtained from the literature or identified as highly differentially expressed. Heatmaps were generated using the heatmap.2 function from the gplots v3.5.2 R package using the default complete-linkage clustering algorithm (*Xu et al., 2021*).

## Pseudotime trajectory analysis

Pseudotime trajectory was plotted by using the R package monocle Version 2.22.0 with the default settings given there. Pseudotime ordering was performed using the function 'reduce dimension' with

max_components set at 2 and reduction_method set as DDRTree. Next, the significantly affected genes were obtained from the top 50 markers among the clusters by using the function differential-GeneTest (fullModelFormulaStr = ~Pseudotime) and were plotted with the function plot_pseudo-time_heatmap. The num_cluster was set at 3 to obtain three modules of significantly changed genes that had similar trends according to their pseudotemporal expression patterns.

## RNA velocity analysis

Cell RNA velocity analysis was performed using the velocyto (V0.17.17) program (*La Manno et al., 2018*). This approach uses the relative proportions of unspliced and spliced mRNA abundance to provide insights into the anticipated future state of cells. Initial annotation of spliced and unspliced reads was performed through the utilization of the 'velocyto.py' commandline tools. Subsequently, downstream analysis was performed using the same 'velocyto.py' framework. The resulting RNA velocity data represents a vector that predicts the transition dynamics of individual cells, with the direction and speed of each transition inferred from the amplitude and direction of the individual cell velocity arrows plotted on a t-Distributed Stochastic Neighbor Embedding (t-SNE) plot. By examining the directional flow within the RNA velocity vector field, the hierarchical relationship between two cell populations can be inferred.

## Pathway enrichment analysis

Functional enrichment analyses, including Gene Ontology (GO) and Kyoto Encyclopedia of Genes and Genomes (KEGG), were conducted to determine which differentially expressed genes (DEGs) significantly enriched GO terms or metabolic pathways. Given that GO is an international standard classification system for gene function, DEGs are mapped to the GO terms (biological functions) in the database. The number of genes in each term was calculated, and a hypergeometric test was performed to identify significantly enriched GO terms in the gene list out of the background of the reference gene list. GO terms and KEGG pathways with false discovery rates $p < 0.05$ were considered significantly different.

## Magnetic-activated cell sorting (MACS) and quantitative real-time PCR

MACS was employed to isolate T and B cells from kidney leukocytes in both SVCV-infected and PBS-administered (control) zebrafish. In this procedure, leukocytes were initially blocked with 5% normal goat serum for 15 min at 10°C, then incubated with rabbit anti-TCRα/β (1:200, produced in our previous studies) or mouse anti-IgM Ab (1:200, produced in our previous studies) for an additional 15 min at 10°C (*Wan et al., 2016*; *Zhu et al., 2014*). Subsequent washing with MACS buffer (PBS containing 2 mM EDTA and 0.5% BSA) was followed by a 15 min incubation at 10°C with anti-IgG magnetic beads (Miltenyi Biotec). The cell suspension was introduced to a LS separation column following the manufacturer's instructions. Total RNA was extracted from positive cells, kidney, and spleen tissues using an RNAiso Plus kit (Takara Bio), and subsequently reverse-transcribed into cDNAs. Quantitative real-time PCR (RT-qPCR) was employed to analyze the transcript abundance of target genes on a CFX Connect Real-Time PCR Detection System (Bio-Rad). The PCR protocol encompassed a total volume of 10 µl with iTaq Universal SYBR Green Supermix (Bio-Rad). Reaction mixtures underwent an initial incubation at 95 °C for 2 min, succeeded by 40 cycles of 15 s at 95 °C, 15 s at 60 °C, and 20 s at 72 °C. Relative expression levels were calculated using the $2^{-\Delta\Delta CT}$ method, with β-actin as the reference gene. Each PCR trial comprised triplicate parallel reactions, and each experiment was independently repeated at least three times. The RT-qPCR primer sequences are detailed in *Supplementary file 9*.

## ScRNA-Seq differential expression analysis and SVCV reads enrichment

Seurat package FindMarkers function with default parameters was used to perform differential gene expression analysis between the control and treatment groups of the same cell type. Within the FindAllMarkers function, the Wilcoxon rank-sum test was applied to assess single-cell differential gene expression for identified cell subsets. Marker genes for each cluster were determined by contrasting them with all other cells in the experiment. Differential gene expression, reflecting diverse levels of upregulation and downregulation in different comparisons, was depicted through volcano plots. Enrichment of reads across SVCV genes was analyzed using the Integrative Genome Viewer to identify

read pileups (*Robinson et al., 2011*; *Speranza et al., 2021*). Cells containing any counts of viral genes were categorized as positive for viral RNA.

## Flow cytometry (FCM) analysis

For FCM analysis, cells underwent blocking with 5% normal goat serum at 4 °C for 1 hr, followed by incubation with corresponding primary antibodies: rabbit anti-CD4-1 (1:200), rabbit anti-CD8α (1:200), and mouse anti-CD154 (1:200), at 4 °C for 2 hr. As a control, non-specific mouse or rabbit IgG served as an isotype control. Subsequently, cells were washed and further incubated with secondary antibodies: PE-conjugated goat anti-mouse IgG and FITC-conjugated goat anti-rabbit IgG, for 1 hr at 4 °C. Fluorescence signals were detected using a FACScan flow cytometer (FACSCalibur, BD Biosciences) at 488 nm. FCM analysis was based on forward/side scatter (FSC/SSC) characteristics and PE/FITC-conjugated fluorescence with CellQuest program as previously described (*Wan et al., 2016*; *Shi et al., 2019*; *Shao et al., 2018*). A minimum of 10,000 events were collected from the lymphocyte gate. Data processing was executed using FlowJo 7.6 software (BD Biosciences). To examine CD4$^+$ and CD8$^+$ T cell proliferation, zebrafish under PBS/inactivated SVCV treatment received intraperitoneal injections of 5-ethynyl-2'-deoxyuridine, Alexa Fluor 488 (EdU, Beyotime, 8 µg/fish) twice within a 24 hr interval two days before euthanization. Seven days after SVCV stimulation, leukocytes from the kidney were collected and incubated with rabbit anti-CD4/CD8 (1:500), followed by APC-conjugated rabbit anti-IgG secondary antibody (1:1000; Thermo Fisher, A-10931). Subsequently, cells were fixed with 4% paraformaldehyde, permeabilized with 0.3% Triton X-100 in PBS, and underwent click reaction according to the manufacturer's instructions (BeyoClick EdU Cell Proliferation Kit with Alexa Fluor 488, Beyotime). The proliferation of CD4$^+$ and CD8$^+$ T cells was subsequently analyzed using FCM (FACSCalibur, BD Biosciences) (*Hong et al., 2023*).

## Immunofluorescence staining

For kidney staining, cryosections were prepared and fixed for 10 min in 95% ethyl alcohol, followed by incubation with primary antibodies. Background autofluorescence was eradicated by treating cryosections with 0.1 M glycine (pH 2.3) for 10 minutes. After washing, kidney tissues were incubated with primary antibodies: rabbit anti-CD4-1 (1:200), rabbit anti-CD8α (1:200), and mouse anti-IgM (1:200), at 4 °C for 2 hr. A control was established using non-specific mouse or rabbit IgG as an isotype control. Subsequently, cryosections were washed and further incubated with secondary antibodies: APC-conjugated goat anti-rabbit/mouse IgG (Thermo Fisher, A-10931, A-865) for 1 hour at 4 °C. The cryosections were then permeabilized using 0.3% Triton X-100 in PBS, and underwent click reaction according to the manufacturer's instructions (BeyoClick EdU Cell Proliferation Kit with Alexa Fluor 488, Beyotime). Prior to photomicrography, additional staining with 4',6-diamidino-2-phenylindole (DAPI, 100 ng/ml; Sigma-Aldrich) was performed. Samples were captured utilizing a Laser Scanning Confocal Microscope (Olympus FV3000) with a 60x oil-immersion objective lens (*Wan et al., 2016*; *Hong et al., 2023*).

## High-throughput IgM antibody repertoire analysis

Zebrafish were intraperitoneally immunized with TNP-KLH plus CpG-ODNs or CpG-ODNs alone (as a control). TNP-KLH (Biosearch Technologies, T5060) was dissolved in PBS and administered at a final dose of 2.5 µg per fish. CpG-ODNs, specifically CpG-2007 (TCGTCGTTGTCGTTTTGTCGTT) and CpG-D (ACCGATAACGTTGCCAACGTTGGT), were synthesized with a phosphorothioate modification by Generay Biotechnology Company (Shanghai, China). These two CpG-ODNs were dissolved in TE buffer (10 mM Tris-HCl, 1 mM EDTA, pH 8.0), mixed in a 1:1 ratio, and administered at a final dose of 1 µg per fish. After 14 days of immunization, cDNA samples extracted from the kidneys of three representative fish were subjected to individual amplification. This was achieved by utilizing specific forward primers corresponding to distinct subgroups of IGHV genes, combined with reverse primers that are specific to the IGHM gene (Cµ) (*Weinstein et al., 2009*). After the PCR reaction, the PCR product was mixed with 2 µl of loading buffer and loaded into a 1.0% agarose gel stained for visualization. Concurrently, 3 µl aliquots of all PCR products obtained from the same tissue were pooled together for subsequent RNaseq analysis. DNA concentrations were measured using the QuBit DNA quantification system (Invitrogen) and the quality checked using an Agilent 2100 Bioanalyzer (Agilent Technologies). A library was constructed per individual tissue with the TruSeq DNA PCR-Free Library

Prep Kit (Illumina) according to the manufacturer's instructions. Libraries were pooled together and paired-end sequencing was performed on an Illumina MiSeq with a MiSeq Reagent Kit v3 (2×300 cycles) cartridge (Illumina). The primers are listed in *Supplementary file 9*.

## Statistical analysis

Statistical analysis and graphical presentation were performed with GraphPad Prism and Origin Pro. All data were presented as the mean ± standard deviation (SD) of each group. Statistical evaluation of differences was assessed using one-way ANOVA, followed by an unpaired two-tailed t-test. Statistical significance was defined as $^*p<0.05$, $^{**}p<0.01$, $^{***}p<0.001$. All experiments were replicated at least three times.

## Acknowledgements

This work was supported by grants from the National Natural Science Foundation of China, and the National Key Research and Development Program of China.

## Additional information

### Funding

| Funder | Grant reference number | Author |
|---|---|---|
| National Natural Science Foundation of China | 32173003 | Li-xin Xiang |
| National Natural Science Foundation of China | 31630083 | Jian-zhong Shao |
| National Key Research and Development Program of China | 2018YFD0900503 | Jian-zhong Shao |
| National Key Research and Development Program of China | 2018YFD0900505 | Li-xin Xiang |

The funders had no role in study design, data collection and interpretation, or the decision to submit the work for publication.

### Author contributions

Chongbin Hu, Software, Formal analysis, Supervision, Validation, Investigation, Visualization, Methodology, Writing - original draft, Writing - review and editing; Nan Zhang, Formal analysis, Investigation, Visualization, Methodology, Writing - review and editing; Yun Hong, Formal analysis, Investigation, Methodology, Writing - review and editing; Ruxiu Tie, Provided consultation, reviewed and revised the manuscript; Dongdong Fan, Provided material support; Aifu Lin, Provided consultation, reviewed and revised the manuscript; Ye Chen, Supervision, Methodology, Provided consultation, reviewed and revised the manuscript; Li-xin Xiang, Supervision, Funding acquisition, Writing - review and editing; Jian-zhong Shao, Conceptualization, Data curation, Formal analysis, Supervision, Funding acquisition, Investigation, Visualization, Methodology, Writing - original draft, Project administration, Writing - review and editing

### Author ORCIDs

Jian-zhong Shao http://orcid.org/0000-0003-3483-1817

### Ethics

All animal care and experimental procedures were approved by the Committee on Animal Care and Use and the Committee on the Ethics of Animal Experiments of Zhejiang University (ZU_0325/2021; Hangzhou, China).

Reviewer #1 (Public Review): https://doi.org/10.7554/eLife.92424.3.sa1

Reviewer #2 (Public Review): https://doi.org/10.7554/eLife.92424.3.sa2
Author Response https://doi.org/10.7554/eLife.92424.3.sa3

## Additional files

### Supplementary files

- Supplementary file 1. Gene sets for all immune cell and HSPCs annotation.

- Supplementary file 2. Marker gene list and gene expression tables of 13 immune-cell and renal cell categories.

- Supplementary file 3. Marker gene list and gene expression tables of HSPC subsets.

- Supplementary file 4. Marker gene list and gene expression tables of T/NK cell subsets.

- Supplementary file 5. Marker gene list and gene expression tables of B cell subsets.

- Supplementary file 6. Marker gene list and gene expression tables of macrophages/myeloid cell subsets.

- Supplementary file 7. Marker gene list and gene expression tables of neutrophil subsets.

- Supplementary file 8. Similarities and differences in HSPC and immune cell populations between zebrafish and human.

- Supplementary file 9. Primers used in this study for Real-time quantitative PCR and IgM antibody repertoire analysis.

- MDAR checklist

### Data availability

The data that support the findings of this study are openly available in GEO at https://www.ncbi.nlm.nih.gov/geo/, GEO accession number GSE242133.

The following dataset was generated:

| Author(s) | Year | Dataset title | Dataset URL | Database and Identifier |
|---|---|---|---|---|
| Hu C | 2024 | Single-cell RNA sequencing unveils the hidden powers of zebrafish kidney for generating both hematopoiesis and adaptive antiviral immunity | https://www.ncbi.nlm.nih.gov/geo/query/acc.cgi?acc=GSE242133 | NCBI Gene Expression Omnibus, GSE242133 |

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
