## [Editor Report · eLife assessment]

This study characterizes the composition and immune diversity of the zebrafish kidney, the immune organ equivalent to human bone marrow, with **convincing** single-cell transcriptomic data of hematopoietic cells and immunocytes. The key findings suggest that zebrafish kidney is a secondary lymphatic organ, and that hematopoietic stem cells in zebrafish may exhibit trained immunity, which are the unique features of the fish immune system. This study provides new and **valuable** insights into the antiviral response in teleost fish, which will be of interest to biologists in general, and to immunologists and cancer researchers in particular.

---

## [Referee Report · Reviewer #1 (Public Review)]

Hu et al. performed sc-RNA-seq analyses of kidney cells with or without virus infection, vaccines, and vaccines+virus infections from pooled adult zebrafish. They compared within these experimental groups as well as kidney vs spleen. Their analyses identified expected populations but also revealed new hematopoietic stem/progenitor cell (HSPC), even in spleen. Their analyses show that HSPCs in kidney can respond to virus infection differentially and can be trained to recognize the same infection and argue that zebrafish kidney can serve as a secondary immune organ. The findings are important and interesting. The manuscript is well written and a pleasure to read.

---

## [Referee Report · Reviewer #2 (Public Review)]

Summary:

In this manuscript, the authors have meticulously constructed a comprehensive atlas delineating hematopoietic stem/progenitor cell (HSPC) and immune-cell types within the zebrafish kidney, employing single-cell transcriptome profiling analysis. Notably, these cell populations exhibited distinctive responses to viral infection. Intriguingly, the investigation revealed that HSPCs manifest positive reactivities to viral infection, indicating the effective induction of trained immunity in select HSPCs. Furthermore, the study unveiled the capacity for the generation of antigen-stimulated adaptive immunity within the kidney, suggesting a role for the zebrafish kidney as a secondary lymphoid organ. This research elucidates the distinctive features of the fish immune system and underscores the multifaceted biology of the kidney in ancient vertebrates.

Strengths:

This study, encompassing 13 figures along with supplementary material, distinguishes itself as one of the most comprehensive investigations on this subject to date.

---

## [Author Response]

The following is the authors’ response to the original reviews.

**eLife assessment**
This manuscript presented convincing single-cell transcriptomic data of hematopoietic cells and immunocytes in zebrafish kidney marrow and showed that these cells have distinctive responses to viral infection. The findings in this study suggest that zebrafish kidney is a secondary lymphatic organ and hematopoietic stem cells in zebrafish may exhibit trained immunity. This represents a valuable discovery of the unique features of the fish immune system.
**Public Reviews:**

**Reviewer #1 (Public Review):**
Hu et al. performed sc-RNA-seq analyses of kidney cells with or without virus infection, vaccines, and vaccines+virus infections from pooled adult zebrafish. They compared within these experimental groups as well as kidney vs spleen. Their analyses identified expected populations but also revealed new hematopoietic stem/progenitor cell (HSPC), even in the spleen. Their analyses show that HSPCs in the kidney can respond to virus infection differentially and can be trained to recognize the same infection and argue that zebrafish kidney can serve as a secondary immune organ. The findings are important and interesting. The manuscript is well written and a pleasure to read. However, there are several issues with their figure presentation and figure qualities, as well as the lack of clarity in some of figure legends. Some of the data presentation can be improved for better clarity. It is also important to outline what is conserved and what is unique for fish.Major concerns:(1) The visualization for several figure panels is very poor. Please provide high resolution images and larger font sizes for gene list or Y and X axis labels. This includes Figure 1B, Figure 1-figure supplement 2, Figure 2B-2C, 3A-3D, 4F, 5B, 6G, Figure 6-figure supplement 1B, Figure 6-figure supplement 2. Figure 7B, 8C-8E, Figure 8-figure supplement 1., 10F, 10G-10J, Figure 10-figure supplement 1.

Response: We apologize for the issue you have pointed out concerning the inadequate visualization of the graphic panels. It is likely that the formatting of the inserted images was altered during the manuscript upload process, leading to a reduction in resolution. However, the graphics uploaded as separate image files, specifically formatted as vector files in PDF format, preserve their high resolution even when zoomed in. Therefore, we kindly request the reviewer to consult the figures in the submission folder for a more detailed examination. We sincerely apologize for any inconvenience caused.

(2) What are the figures at the end of the manuscript without any figure legends?

Response: Thank you for bringing this issue to our attention. The last few figures that lack figure legends are actually supplementary figures included in the text. It is possible that they were automatically and repeatedly generated by the submission system. In the revised manuscript, we will take measures to ensure that this issue is avoided.

(3) It would be better to use a Table to organize the gene signatures that define each unique population of immune cells such as T, B, NK, etc.

Response: We greatly appreciate the valuable advice provided by the reviewer. As per the reviewer's recommendation, we have included a comprehensive display of all cell types and corresponding gene signatures in Supplementary File 1 of the revised manuscript.

(4) What are the similarities for HSPC and immune cell populations between fish and man based on this research? It is better to form a table to compare and discuss.

Response: Following the valuable suggestion of the reviewer, we have included an additional comparative analysis of HSPC and immune cell populations between zebrafish and humans. This information can be found in Supplementary file 8 and in the "Discussion" section (lines 684-685).

(5) It is highly likely that sex and age could be the biological variation for how HSPC responds to virus infections and vaccination. The author should clearly state the fish sex and age from their samples and discuss their results taking into consideration of these variations.

Response: We are grateful for the reviewer's insightful comments. To reduce inter-individual variations, zebrafish samples were selected randomly, with an equal distribution of males and females, during their prime youth period spanning from 3 to 12 months of age. We have included supplementary instructions regarding this selection process in the "Materials and Methods" section (lines 798-799).

(6) The authors claim that the spleen and kidney share HSPCs. However, their data did not demonstrate this result clearly in Figure 4A. Perhaps they should use different color to make the overlay becoming more obvious? Or include a table to show which HSPCs are shared between the kidney and spleen? Are they sure if these are just HSPCs seeding the spleen to differentiate into B cells or other immune cells?

Response: We express our gratitude to the reviewer for raising this issue. In this section, we would like to provide detailed explanations regarding this matter. It is important to note that the figures positioned on both the left and right sides of Figure 4A should be interpreted in a corresponding manner. The left-side figure represents the cellular composition from the spleen (depicted in light red) and the kidney (depicted in blue) across various cell types. Each data point in the left-side figure signifies an individual cell, with the two distinct colors indicating the origin of the cell. On the other hand, the right-side figure displays the varied colors representing different cell types. We want to emphasize that the spatial distribution and proportions of diverse cells in the tSNE plot on the right align consistently with the information presented in the left-side figure. This indicates the correspondence between the two plots and reinforces the validity of our findings. When interpreting the figures on the left and right sides of Figure 4A in a corresponding manner, it becomes evident that the overlapping HSPCs shared by both spleen and kidney predominantly reside in the HSPCs1 group (indicated as cluster 5 in the right-side figure). Additionally, there is also a small distribution of the overlapping HSPCs in the HSPCs2 group (cluster 8 in the right-side figure). These observations underline the presence of overlapping HSPCs in both the kidney and spleen. However, further clarification is required to fully comprehend the intricate correlation between the HSPCs in the kidney and spleen.

**Reviewer #1 (Recommendations For The Authors):**
Minor concerns:(1) Figure 3C: why is 10 listed in between 1 and 2?

Response: We appreciate the reviewer's comment. It is pertinent to mention that the graphs in Figure 3C underwent an automatic sorting process facilitated by the software during the analysis. It should be emphasized that the assigned positions resulting from this sorting process have no bearing on the outcomes of the analysis.

(2) Figure 4A: difficult to assess the overlay between the kidney and spleen.

Response: As mentioned above, the overlapping HSPCs shared by both the spleen and kidney are mainly distributed in the HSPCs1 group (cluster 5 in the right-side figure), with a small amount also found in the HSPCs2 group (cluster 8 in the right-side figure).

(3) Figure 4C: What is this sample, kidney or spleen? Please specify.

Response: Figure 4C represents an overlay of the spleen and kidney cells depicted in Figure 4B, which includes all cells of the spleen and kidney to show the differentiation trajectory of the cells. As per reviewer’s suggestion, we have made corresponding modification to the revised figure.

(4) The manuscript is very long. Consider to focus on the major findings as the main figures and move the rest to the supplementary figures.

Response: This article aimed to comprehensively understand the hematopoietic and immunological traits of zebrafish kidneys through a systematic study. As a result, a comprehensive presentation of the findings has been provided. Given that the figures currently integrated into the main text play a significant role in illustrating the principal outcomes of each section, we kindly request that these figures remain in the main body of the article. This will contribute to sustaining the structural coherence and readability of the manuscript. Thank you for taking our request into consideration.

**Reviewer #2 (Public Review):**
In this manuscript, the authors have meticulously constructed a comprehensive atlas delineating hematopoietic stem/progenitor cell (HSPC) and immune-cell types within the zebrafish kidney, employing single-cell transcriptome profiling analysis. Notably, these cell populations exhibited distinctive responses to viral infection. Intriguingly, the investigation revealed that HSPCs manifest positive reactivities to viral infection, indicating the effective induction of trained immunity in select HSPCs. Furthermore, the study unveiled the capacity for the generation of antigen-stimulated adaptive immunity within the kidney, suggesting a role for the zebrafish kidney as a secondary lymphoid organ. This research elucidates the distinctive features of the fish immune system and underscores the multifaceted biology of the kidney in ancient vertebrates.

Response: We would like to express our gratitude to the reviewers for their overall positive feedback on our article.

**Reviewer #2 (Recommendations For The Authors):**
(1) The authors propose that zebrafish kidney is a dual-functional entity with functionalities of both primary and secondary lymphoid organs. Do the authors have any insights into the coordination of these two functions in the kidneys?

Response: We are grateful for the valuable comments provided. We believe that the question raised by the reviewer poses an intriguing research topic, as it explores the intricate interaction between the hematopoietic and adaptive immune systems in the renal organ. This exploration holds significant value in understanding the underlying mechanisms. To accomplish this, advanced techniques such as spatiotemporal single-cell transcriptomics and dynamic cell tracking will be utilized to validate the interplay between hematopoietic and immune cell lineages.

(2) Previous studies have found that fish IgZ/IgT specificity exists in mucosal immune organs. Is the expression of the zebrafish IgZ gene observed in the kidney? If so, is there any correlation with IgZ in mucosal immune organs?

Response: Thank you for drawing attention to this matter. In our study, we observed the expression of the IgZ gene (ighz) in the zebrafish kidney, as shown in Figure 6. This discovery aligns with previous research and confirms its presence in B cells. While IgZ is known to function as an antibody in mucosal immunity, it remains unclear whether the development of its secretory cells (IgZ+ B cells) originates from the central immune system, such as the kidney. Our results suggest that IgZ+ B cells may have their origin in the kidney and then migrate through the peripheral circulation to carry out their functions in the local mucosal system. This finding is consistent with our earlier research, which demonstrated that zebrafish IgZ is not limited to mucosal immune organs but is also abundantly present in systemic immunity, including peripheral blood (Immunology. 2021; 162(1): 105-120).

Reference:

Ji, J. F. et al. Differential immune responses of immunoglobulin Z subclass members in antibacterial immunity in a zebrafish model. Immunology, 2021;162(1), 105-120.

(3) Did the authors use the zebrafish genome or transcriptome for gene annotation? If the former, which version is used? Please supplement in the "Materials and methods".

Response: We appreciate the comments provided by the reviewer. In this study, we utilized the zebrafish genome, specifically the GRCz11 version, to annotate genes. The detailed genome data can be found at http://asia.ensembl.org/Danio_rerio/Info/Index. We have incorporated this information into the "Materials and Methods" section of the revised manuscript (line 873).

(4) Since the authors performed single-cell sequencing on leukocytes, why did several kidney cells, such as kidney multicellular cells and kidney mucin cells existed in the samples?

Response: Thanks for the reviewer’s comments. It is important to acknowledge that inadvertent mixing of kidney cells might have occurred during the preparation of single-cell suspensions in our analyzed sample. However, it is pertinent to emphasize that our primary focus was the analysis of immune cells. Therefore, any minor contamination from kidney cells in the analyzed sample is considered negligible and does not significantly affect the main results of our analysis.

(5) The application of "trained immunity," although currently popular, appears unsuitable in this context, as the current scenario involves a recall with the cognate antigen.

Response: To our knowledge, trained immunity is generally recognized as the long-term memory of innate immunity based on transcriptional, epigenetic and metabolic modifications of myeloid cells, which are characterized by elevated pro-inflammatory responses to secondary stimuli, whether they are identical or different (Cell Host Microbe. 2012; 12(2): 223-32; Nat Immunol. 2021; 22(1): 2-6; J Clin Invest. 2022;132(7): e158468). Therefore, stimulation of cognate antigens can be considered as a form of training immunity, and we hope that it will be accepted in this context.

References:

(1) Quintin, J. et al. Candida albicans infection affords protection against reinfection via functional reprogramming of monocytes. Cell host & microbe, 2012;12(2), 223-232.

(2) Divangahi, M. et al. Trained immunity, tolerance, priming and differentiation: distinct immunological processes. Nature immunology, 2021;22(1), 2-6.

(3) Pernet, E. et al. Training can’t always lead to Olympic macrophages. Journal of Clinical Investigation, 2022;132(7), e158468.

(6) The discovery that HSPC exhibits trained immune characteristics is novel. Do the authors have any insights into the biological significance of trained immunity in HSPCs concerning immune defense?

Response: We propose that the generation of trained immunity in HSPCs holds significant physiological implications. This process may expedite the differentiation and activation of specific immune cells upon re-infection, thereby bolstering the body's immune defenses and pathogen clearance. Consequently, it may serve as an intelligent strategy for host defense against pathogens. However, additional research is required to confirm this hypothesis.

(7) In the Figure 13I, the authors used CpG and CpG+TNP-KLH to stimulate zebrafish, but no corresponding experimental method was provided in the "Materials and methods". Please supplement.

Response: Thanks for the reviewer’s careful reading. We have included corresponding supplementary instructions in the “Materials and methods” section (lines 1011-1018).

(8) At line 187-190 in "Results", authors state that "It's noteworthy that cluster 11 exhibited high expression of genes ......, resembling a unique serpin-secreting cell population". Noteworthy is the fact that serpins play a role in diverse immunological processes, including coagulation, inflammation, as well as myeloid and lymphoid cell development. Could this renal cell cluster (kidney mucin cells) potentially harbor immunological functions?

Response: Given the crucial role of serpins in various immunological processes, secreted serpins from this particular cell cluster likely possess significant immunological functions, suggesting the notable immunological capabilities of this cell group. Consequently, our forthcoming research aims to conduct a more comprehensive investigation of this specific cell population.

(9) At line 171 in "Results", the number "6" in the "cluster 6" should not be italicized, please correct.

Response: We have addressed this issue in the revised manuscript (line 170).

(10) At line 937 in "Materials and methods", the authors isolated T/B lymphocytes through magnetic bead sorting. Please provide information on the source of the antibodies (rabbit anti-TCRα/β or mouse anti-IgM Ab).

Response: We have included corresponding instructions in the “Materials and methods” section (lines 938-939).